# MONOPHILIC NEIGHBOURHOOD TRANSFORMERS

## ABSTRACT

Graph neural networks (GNNs) have seen widespread application across diverse fields, including social network analysis, chemical research, and computer vision. Nevertheless, their efficacy is compromised by an inherent reliance on the homophily assumption, which posits that adjacent nodes should exhibit relevance or similarity. This assumption becomes a limitation when dealing with heterophilic graphs, where it is more common for dissimilar nodes to be connected. Addressing this challenge, recent research indicates that real-world graphs generally exhibit monophily, a characteristic where a node tends to be related to the neighbours of its neighbours. Inspired by this insight, we introduce Neighbourhood Transformers (NT), a novel approach that employs self-attention within every neighbourhood of the graph to generate informative messages for the nodes within, as opposed to the central node in conventional GNN frameworks. We develop a neighbourhood partitioning strategy equipped with switchable attentions, significantly reducing space consumption by over 95% and time consumption by up to 92.67% in NT. Experimental results on node classification tasks across 5 heterophilic and 5 homophilic graphs demonstrate that NT outperforms current state-of-the-art methods, showcasing their strong performance and adaptability to different graph types. The code for this study is available at `https://anonymous.4open.science/r/MoNT-BD3C`.

## 1 INTRODUCTION

Graph neural networks (GNNs) have emerged as a fundamental technology in the realm of graph learning, garnering extensive research interest and a wealth of practical applications over the past decade (Wu et al., 2021b). Their versatility has been demonstrated across a wide array of disciplines. In the domain of social network analysis, for instance, GNNs are utilized to predict user interactions and to pinpoint pivotal influencers within the network (Fan et al., 2019). Within the field of chemistry, GNNs are instrumental in predicting molecular attributes and unravelling the mechanisms behind chemical reactions (Gilmer et al., 2017). In the realm of computer vision, GNNs are applied to model the complex interdependencies among visual components, enhancing tasks such as object detection and scene graph generation (Sarlin et al., 2019). Central to the functionality of GNNs is the message passing mechanism, which facilitates the exchange of information between nodes and their adjacent neighbours, thereby allowing GNNs to harness both node features and the structural topology of the graph (Scarselli et al., 2009).

Message passing (MP) implicitly posits that adjacent nodes are relevant or similar to one another, as is often the case in social networks where connected individuals tend to share similar interests (McPherson et al., 2001; Gerber et al., 2013; Ciotti et al., 2015). However, recent investigations have called into question this homophily assumption by introducing a collection of heterophilic benchmark graphs, where the premise of similarity between neighbouring nodes does not consistently apply (Pei et al., 2020; Lim et al., 2021; Platonov et al., 2023). For instance, in financial transaction networks, the majority of users with whom fraudsters engage in transactions are not engaged in fraudulent activities themselves (Pandit et al., 2007). On such heterophilic graphs, researchers have noted that a node often exhibits similarity not with its immediate neighbours, but with its 2-hop neighbours, or the neighbours of its neighbours (Zheng et al., 2022; Zhu et al., 2020). This characteristic, referred to as monophily (Altenburger & Ugander, 2018; Chin et al., 2019), is also prevalent in homophilic graphs (Lei et al., 2022; Xiao et al., 2023). Consequently, the monophily assumption appears to be a universal trait in graphs, irrespective of their degree of homophily.

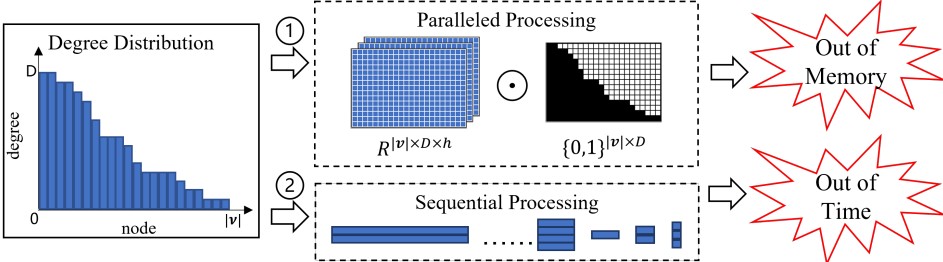

Figure 1: Unevenly distributed neighbourhood sizes. ① **Paralleled Processing** pads node features of all neighbourhoods and processes the padded tensor in a single operation, occupying excessive memory when the node degree distribution is long-tailed. ② **Sequential Processing** organizes neighbourhoods by size and processes node features group by group, consuming prohibitive time when the node degree distribution is dispersed.

Drawing upon the monophily assumption, we introduce Neighbourhood Transformers (NT), which enables message exchanging among nodes within each neighbourhood through self-attention (Vaswani et al., 2017) and constructs node representations by aggregating the exchanged messages from the neighbourhoods of all its neighbours. A pivotal challenge of applying self-attention in each neighbourhoods is the high complexity resulting from the variable sizes of neighbourhoods. In real-world graphs, the distribution of node degrees, correspond to the neighbourhood sizes, tends to be scattered and follow a long-tailed pattern, characterized by a small number of nodes with high degrees and a large number of nodes with low degrees (Yin et al., 2012). This attribute presents NT with a quandary, as illustrated in Figure 1: either pad an excessive amount of redundant space to facilitate parallel processing (Vaswani et al., 2017) or tolerate considerable time overhead to implement sequential processing (Yan et al., 2020). Moreover, the quadratic complexity of self-attention must also be taken into account when processing large neighbourhoods. To tackle these challenges, we incorporate linear-attention (Tay et al., 2023) and devise a neighbourhoods partitioning strategy, significantly diminishing both the space and time requirements in NT. For instance, when applied to the Tolokers dataset (Platonov et al., 2023), our method reduces the memory footprint from over 80GB, necessitated by parallel processing, to less than 4GB and accomplishes the training in only 7.33% of the time required for sequential processing. These optimizations render NT practical, enabling us to conduct extensive experiments and assess its superiority against state-of-the-art baselines.

In summary, our contributions include **1)** a model, Neighbourhood Transformers, designed to harness the recently identified property of monophily within real-world graphs, **2)** a neighbourhood partitioning strategy equipped with switchable attentions to reduce space and time consumtions of NT, and **3)** extensive experiments across diverse benchmark graphs and thorough ablation studies.

## 2 RELATED WORKS

**Heterophlilic GNNs** (HGNN) are improved GNNs to address the challenges posed by heterophily (Pei et al., 2020; Maurya et al., 2022; Li et al., 2022; Bo et al., 2021; Du et al., 2022; Wang & Zhang, 2022). The two most prevalent strategies of HGNN are passing messages along the second-order adjacency matrices $A^2$ (Lei et al., 2022; Zhu et al., 2020; Xiao et al., 2023), and aggregating neighbour-embeddings in separation with the ego-embeddings (Zhu et al., 2020; Platonov et al., 2023). Similar to their utilization of $A^2$, NT also accesses information from 2-hop neighbours for a node due to the message exchanging within its belonging neighbourhoods. The key distinction is that the message exchanging in each neighbourhood is uniquely conditioned on its central node, as depicted in Figure 2c, thereby enriching the diversity of exchanging patterns and enhancing the attentiveness of aggregated representations. This flexible conditioning on neighbourhoods' centers of NT also ensures compatibility with traditional message passing and maintains performance in homophilic settings, if disregarding the neighbourhood and focusing on the centers. Besides, the ego-neighbour separation technique of HGNN is not essential for NT, as NT inherently aggregates information from monophilic nodes with similarity, thus obviating the need for such separation.

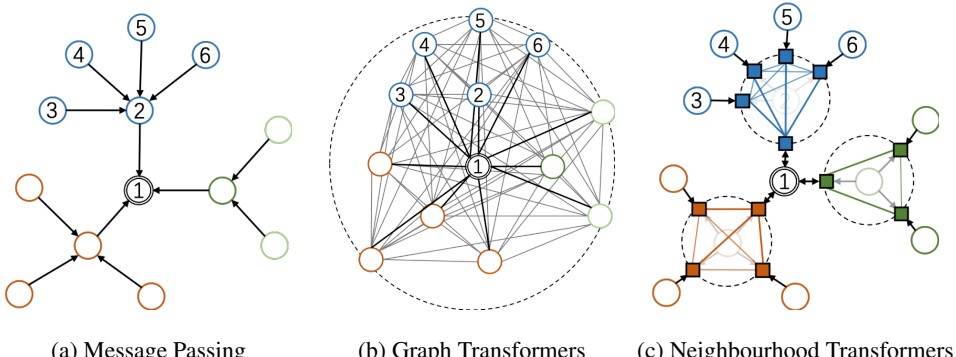

|  (a) Message Passing | (b) Graph Transformers | (c) Neighbourhood Transformers |

Figure 2: An illustration of computing representations for node $v_1$ (represented by a double-lined circle) using various mechanisms. **(a) Message Passing**: Messages from different nodes (e.g., $v_3, v_4, v_5, v_6$) are propagated along edges towards node $v_1$ to compute its representation, but are diluted and over-squashed when passing through heterophilic bottlenecks (e.g., $v_2$). **(b) Graph Transformers**: Node $v_1$ aggregates messages from all nodes using self-attention. This approach may inadvertently downplay the importance of information from nearby nodes due to a reduction in the influence of topological structure. **(c) Neighbourhood Transformers (Ours)**: Node $v_1$ exchanges messages (depicted as squares) in each of its constituent neighbourhoods through self-attention, thereby acquiring attentive and structure-aware representations.

**Graph Transformers** (GT) employ self-attention across the entire node set to capture global data dependencies (Wu et al., 2021a; Chen et al., 2023), as depicted in Figure 2b. This capability is advantageous in overcoming the information bottleneck associated with message passing (MP) (Alon & Yahav, 2021) and in aggregating high-order information to address heterophily challenges (Ying et al., 2021). However, due to the lack of topological regulation, GT often prioritizes distant nodes, potentially overlooking nearby nodes that typically carry more relevant information (Xing et al., 2024). As a trade-off, GT necessitates the explicit integration of structural encodings (Kreuzer et al., 2021; Dwivedi et al., 2022) and the implicit representations of MP to address this shortcoming (Deng et al., 2024). These limitations suggest that GT should be integrated with MP (Shirzad et al., 2023; Ma et al., 2023; Rampásek et al., 2022), rather than replacing MP as a standalone graph learning method. In contrast to GT, Neighbourhood Transformers (NT) apply self-attention within each neighbourhood of the graph and propagate the fully-exchanged messages to all nodes within that neighbourhood, as depicted in Figure 2c. As such, NT maintains the ability to leverage structural information and can focus learning on the 2-hop local structure without succumbing to the issue of over-globalization.

## 3 PRELIMINARIES

We denote a graph as $\mathcal{G} = (\mathcal{V}, \mathcal{E})$, where $\mathcal{V} = \{v_i | i = 1, 2, \ldots, |\mathcal{V}|\}$ is the node set and $\mathcal{E} = \{e_{ij} | v_i \text{ connects } v_j\}$ is the edge set. Node features are a matrix $\boldsymbol{X} \in \mathbb{R}^{|\mathcal{V}| \times d}$ where $d$ is the dimensions of node features. The $i$-th row $\boldsymbol{X}_{i,:}$ corresponds to the feature vector of node $v_i$. Edge attributes are a matrix $\boldsymbol{E} \in \mathbb{R}^{|\mathcal{E}| \times d_e}$ where $d_e$ is the dimensions of edge attributes. We denote the attributes of edge $e_{ij}$ as $\boldsymbol{E}_{(i,j),:}$. The neighbourhood of node $v_i$ is the set of nodes that connect to $v_i$, denoted as $\mathcal{N}(v_i) = \{v_j | e_{ij} \in \mathcal{E}\}$.

### 3.1 MESSAGE PASSING IN GRAPH NEURAL NETWORKS

The predominant architecture of graph neural networks (GNNs) is founded on the message passing (MP) mechanism, which comprises two essential components: the combiner and the aggregator. The combiner is a node-specific function, such as a straightforward linear transformation or a multilayer perceptron (MLP), that encodes the input node features $\boldsymbol{X}$ into messages $\boldsymbol{Z} \in \mathbb{R}^{|\mathcal{V}| \times h}$. The aggregator is an order-invariant operation, like mean (Kipf & Welling, 2016), max (Hamilton et al., 2017), sum (Xu et al., 2019), weighted-mean (Velickovic et al., 2018; Brody et al., 2022), or gated-

sum (Bresson & Laurent, 2017), which is utilized to aggregate the messages $\boldsymbol{Z}$ from neighbouring nodes to produce the final node representations $\boldsymbol{H} \in \mathbb{R}^{|\mathcal{V}| \times h}$. In essence, an MP layer operates as follows:

$$\boldsymbol{Z} = \phi(\text{Combiner}(\boldsymbol{X})), \quad \boldsymbol{H}_{i,:} = \text{Aggregator}(\{\boldsymbol{Z}_{j,:} | v_j \in \mathcal{N}(v_i)\}),$$

where $\phi(\cdot)$ denotes a non-linear activation function, such as GELU (Hendrycks & Gimpel, 2016).

## 3.2 SELF-ATTENTION IN TRANSFORMERS

Self-attention is the core innovation of Transformer, designed to capture intricate dependencies among the $n$ input nodes. It initially maps the node features $\boldsymbol{X} \in \mathbb{R}^{n \times d}$ into corresponding query, key, and value matrices $\boldsymbol{Q}, \boldsymbol{K}, \boldsymbol{V} \in \mathbb{R}^{n \times h}$. Subsequently, an $n \times n$ correlation matrix is generated by the matrix multiplication of $\boldsymbol{Q}$ and $\boldsymbol{K}^T$, which indicates the importance weights for information selection from $\boldsymbol{V}$. Concisely, a self-attention module can be formalized as:

$$(\boldsymbol{Q}, \boldsymbol{K}, \boldsymbol{V}) = \boldsymbol{X} \cdot (\boldsymbol{W}_q, \boldsymbol{W}_k, \boldsymbol{W}_v), \quad \text{SelfAttention}(\boldsymbol{X}) = \text{softmax}(\frac{\boldsymbol{Q} \cdot \boldsymbol{K}^T}{\sqrt{h}}) \cdot \boldsymbol{V},$$

where $\boldsymbol{W}_q, \boldsymbol{W}_k, \boldsymbol{W}_v \in \mathbb{R}^{d \times h}$ are trainable parameters.

## 3.3 LINEAR-ATTENTION IN PERFORMERS

The quadratic complexity $O(n^2)$ of self-attention, as evidenced in the operation $\boldsymbol{Q} \cdot \boldsymbol{K}^T$, becomes computationally infeasible when the number $n$ of nodes is substantial. Consequently, a variety of efficiency-enhanced attention mechanisms with linear complexity $O(n)$ have been proposed (Tay et al., 2023). Among these, Performer (Choromanski et al., 2021) offers an unbiased or nearly-unbiased estimation of self-attention, complete with provable convergence and reduced variance. It initially maps the query and key matrices $\boldsymbol{Q}, \boldsymbol{K}$ into $\hat{\boldsymbol{Q}}, \hat{\boldsymbol{K}} \in \mathbb{R}^{n \times p}$ using its orthogonal random features $\boldsymbol{P} \in \mathbb{R}^{h \times p}$. Next, the product of $\hat{\boldsymbol{K}}^T$ and $\boldsymbol{V}$ results in a $p \times h$ matrix, which then matrix-multiplies $\hat{\boldsymbol{Q}}$ to yield the approximated attention weights. The approximated self-attention mechanism in Performer can be expressed as follows:

$$\hat{\boldsymbol{Q}} = \exp(\frac{1}{\sqrt{h}} \cdot \boldsymbol{Q} \cdot \boldsymbol{P}), \quad \hat{\boldsymbol{K}} = \exp(\boldsymbol{K} \cdot \boldsymbol{P} - \frac{\|\boldsymbol{K}\|^2}{2}),$$

$$\hat{\boldsymbol{D}} = \text{diag}(\hat{\boldsymbol{Q}} \cdot (\hat{\boldsymbol{K}}^T \cdot \mathbf{1}_{n \times 1})), \quad \text{SelfAttention}(\boldsymbol{X}) \approx \hat{\boldsymbol{D}}^{-1} \hat{\boldsymbol{Q}} \cdot (\hat{\boldsymbol{K}}^T \cdot \boldsymbol{V}).$$

When $h$ is held constant and $p$ is set to $h \log h$, as recommended by Performer, the complexity of this linear-attention is $O(nph) = O(n)$.

## 4 METHOD

### 4.1 NEIGHBOURHOOD TRANSFORMERS

Motivated by the universal monophily in graphs, we propose to facilitate the message exchanging among nodes from the same neighbourhoods and implement Neighbourhood Transformer (NT) as:

$$\boldsymbol{Z}_{(j,k),:} = \phi(\text{Combiner}([\boldsymbol{H}'_{j,:} \quad \boldsymbol{H}'_{k,:}])), \tag{1}$$

$$\boldsymbol{M}^{(j)} = \phi(\text{SelfAttention}(\oplus\{\boldsymbol{Z}_{(j,k),:} | v_k \in \mathcal{N}(v_j)\})), \tag{2}$$

$$\boldsymbol{H}_{i,:} = \text{Aggregator}(\{\boldsymbol{M}^{(j)}_{(i),:} | v_j \in \mathcal{N}(v_i)\}), \tag{3}$$

where $\oplus$ denotes the operation of stacking a set of row vectors into a matrix, $\boldsymbol{H}'$ is the node representations from the previous NT layer initialized as $\boldsymbol{H}' = \boldsymbol{X}$. In Equation 3, NT computes the $h$-dimensional representation $\boldsymbol{H}_{i,:}$ for node $v_i$ by aggregating messages $\boldsymbol{M}^{(j)}_{(i),:}$ from all its adjacent nodes $v_j \in \mathcal{N}(v_i)$. Here, $\boldsymbol{M}^{(j)}_{(i),:}$ corresponds to the row vector for node $v_i$ in matrix $\boldsymbol{M}^{(j)}$. The matrix $\boldsymbol{M}^{(j)} \in \mathbb{R}^{|\mathcal{N}(v_j)| \times h}$ encapsulates messages from node $v_j$ to all its neighbours, which are comprehensively exchanged within its neighbourhood $\mathcal{N}(v_j)$ through self-attention, as delineated in Equation 2. This message exchanging exploits the monophily property, a feature that has been

shown to be advantageous in heterophilic graphs, and aids nodes in capturing intricate dependencies among similar 2-hop neighbours. Prior to exchange, in Equation 1, the initial message $\boldsymbol{Z}_{(j,k),:}$ from node $v_j$ to node $v_k$ is computed by combining their node representations $\boldsymbol{H}'_{j,:}$ and $\boldsymbol{H}'_{k,:}$. The message exchanged within each neighbourhood is thus uniquely tailored to its central node, endowing NT with the diversified capability to accommodate various graphs. Besides, the combiner is optimizable, allowing it to adaptively prioritize $\boldsymbol{H}'_{j,:}$ over $\boldsymbol{H}'_{k,:}$ when sole reliance on 1-hop neighbours is sufficient, such as in homophilic graphs, thereby mimicking traditional message passing (MP). We provide the theoretical proofs to this adaptiveness in Appendix E.

## 4.2 Efficient Neighbourhood Transformers

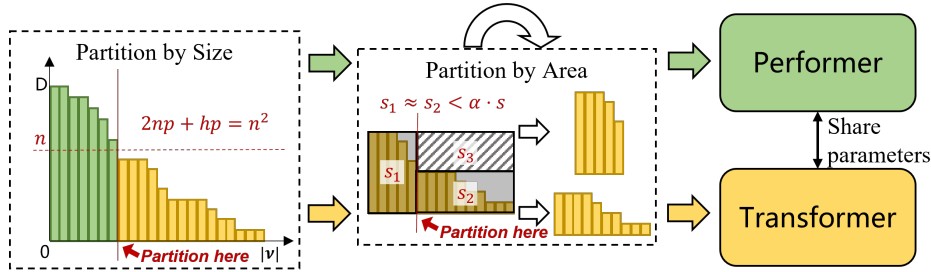

Figure 3: Partitioning neighbourhoods into multiple groups for space and time efficiency. We partition neighbourhoods into two groups based on the degrees of their central nodes. Neighbourhoods with more than $n$ nodes will be processed by Performer for efficiency and other small neighbourhoods will be processed by Transformer for accuracy. Each group of neighbourhoods is recursively partitioned into two approximately equal halves in terms of area ($s_1 \approx s_2$), provided that the partitioning leads to a considerable compression rate $\alpha$.

As we have discussed in Section 1, the primary obstacle to the practical application of NT is the excessive complexity arising from the uneven distribution of neighbourhood scales. To address this challenge, we have developed a neighbourhood partitioning strategy that partitions all neighbourhoods into several smaller groups. Each group is processed through an attention module that can be switched between the precise self-attention of the Transformer and the computationally efficient linear-attention of the Performer. We depict this strategy in Figure 3 and provide a detailed description in the subsequent sections.

### 4.2.1 Partitioning Neighbourhoods by Size

The self-attention mechanism, with its quadratic complexity, necessitates a significant amount of memory when applied to large neighbourhoods. To mitigate this issue, we implement a switchable attention module that processes neighbourhoods with different scales. This module employs the linear-attention of the Performer to handle neighbourhoods exceeding $n$ nodes, prioritizing efficiency, and switches to the self-attention of the Transformer for neighbourhoods with $n$ nodes or fewer, ensuring accuracy. Notably, both attention algorithms share a single set of parameters, which is made possible by the Performer's remarkable property of full compatibility with the Transformer. As detailed in Section 3.3, the self-attention in the Transformer computes an $n \times n$ matrix $\boldsymbol{Q} \cdot \boldsymbol{K}^T$, whereas the linear-attention in the Performer computes two $n \times p$ matrices $\hat{\boldsymbol{Q}}, \hat{\boldsymbol{K}}$ and a $p \times h$ matrix $\hat{\boldsymbol{K}}^T \cdot \boldsymbol{V}$. Here, $p$ and $h$ represent the dimensions of the orthogonal random features in the Performer and the node representations, respectively. Thus, we set $n = p + \sqrt{p^2 + hp}$ by solving the equation $n^2 = 2np + hp$ to ensure that the linear-attention indeed reduces the memory footprint compared to the original self-attention.

### 4.2.2 Partitioning Neighbourhoods by Area

In scenarios where we have $N$ neighbourhoods, and the largest neighbourhood contains $D$ nodes, processing these neighbourhoods in parallel necessitates padding the node features into an $N \times D \times d$ tensor (with an $N \times D$ boolean matrix to indicate the paddings). Due to the long-tail distribution

**Algorithm 1** Partitioning Neighbourhoods by Area

---

**Input:** compression rate $\alpha$, a list $[(n_1, c_1), (n_2, c_2), \ldots, (n_l, c_l)]$ with $n_1 > n_2 > \ldots > n_l$ and
$\sum_{i=1}^{l} c_i = |\mathcal{V}|$ where an element $(n_i, c_i)$ indicates $c_i$ neigbourhoods with size $n_i$.

**Output:** a set $\mathbb{S}$ where an element $(i, j, s)$ indicates a group of neighbourhoods sized in $[n_i, n_j]$ and its area is $s$.

1: **Define:** $\text{Area}(i, j) = (a_j - a_i + c_i) \times n_i$, where $a_i := \sum_{j=1}^{i} c_j$

2: **Initialize:** $\mathbb{S} := \{(1, l, \text{Area}(1, l))\}$

3: **while** true **do**

4:     select $(i, j, s)$ from $\mathbb{S}$ with maximum $s$

5:     **if** $j = i$ **then**

6:         **return** $\mathbb{S}$

7:     **end if**

8:     $t^* := \arg\min_{t=i}^{j} \max(\text{Area}(i, t), \text{Area}(t + 1, j))$

9:     $s_1 := \text{Area}(i, t^*)$

10:    $s_2 := \text{Area}(t^* + 1, j)$

11:    **if** $\max(s_1, s_2) \geq \alpha \cdot s$ **then**

12:        **return** $\mathbb{S}$

13:    **end if**

14:    $\mathbb{S} := \mathbb{S} \setminus \{(i, j, s)\} \cup \{(i, t^*, s_1), (t^* + 1, j, s_2)\}$

15: **end while**

---

characteristic of real-world graphs, this pre-processing step often results in substantial space wastage due to padding and an increase in time due to redundant computation. To address this issue, we propose to partition the neighbourhoods into smaller groups and process them sequentially rather than in a single paralleled operation. As Figure 3 shows, processing the two partitioned groups sequentially results in a memory footprint proportional to the area of $\max(s_1, s_2)$, which is smaller than the memory footprint of the original group, proportional to the area of $s = s_1 + s_2 + s_3$.

Partitioning a group of neighbourhoods into two can reduce not only memory footprint but also redundant computation on padded bits. However, it may increase processing time due to the sequential handling of the partitioned halves. We therefore adopt the partitioning strategy only when the compression rate $\max(s_1, s_2)/s$ is significantly reduced, indicating a substantial enough saving in redundant computation to offset the potential increase in sequential processing.

As a result, we develop an algorithm, described in Algorithm 1, to adaptively partition a group of neighbourhoods into multiple parts for both space and time efficiency. In the algorithm, we define a function to calculate the area of a neighbourhood group as the product of the number of its containing neighbourhoods and their maximum size (in line 1). Then, we initialize the algorithm with all inputted neighbourhoods as a single group (in line 2). In the main loop, we repeatedly select the group with the largest area from the set (in line 4) and partition it into two halves with minimal and approximal areas (in line 8). We prove that, in Appendix F, this partitioning can lead to minimal memory consumption required by neighbourhood transformers. If the partitioning leads to a considerable compression rate (in line 11), we accept the partitioning and replace the group with its two halves (in line 14). By adjusting the hyperparameter $\alpha$, we can control the tradeoff between memory usage and processing time. Our empirical findings suggest that $\alpha = 0.4$ is a good balance, resulting in fast processing with relatively low memory consumption. When a group is atomic (line 6) or a partitioning is refused (line 12), we terminate partitioning other smaller groups and output the group set $\mathbb{S}$.

### 4.3 EXTENSIONS FOR NEIGHBOURHOOD TRANSFORMERS

#### 4.3.1 DYNAMIC AGGREGATORS

In addition to the static aggregators, such as $\text{mean}, \text{max}, \text{sum}$, we harness the exchanged messages to simplify the implementation of dynamic aggregators. Specifically, we double the dimensions of

the output matrices $\boldsymbol{M}^{(j)}$ from Equation 2 and split them into two parts. The first part is normalized by $\sigma$, which corresponds to the Softmax function for the weighted-mean aggregator and the Sigmoid function for the gated-sum aggregator, to generate the weights for aggregating the second part. This results in Equation 3 being expressed as:

$$\boldsymbol{H}_{i,:} = \text{Aggregator}(\{\boldsymbol{M}_{(i),:}^{(j)} | v_j \in \mathcal{N}(v_i)\}) = \sum_{v_j \in \mathcal{N}(v_i)} \sigma(\frac{1}{h} \cdot \sum_{l=1}^{h} \boldsymbol{M}_{(i),l}^{(j)}) \cdot \boldsymbol{M}_{(i),h+1:2h}^{(j)}.$$

The weights of the messages $\boldsymbol{M}_{(i),h+1:2h}^{(j)}$ from node $v_j$ to node $v_i$ are determined not solely by the two endpoints but by the entire neighbourhood $\mathcal{N}(v_j)$, thereby enhancing the attentiveness and suitability of our approach for NT.

### 4.3.2 DIR-NT: DIRECTED NEIGHBOURHOOD TRANSFORMERS

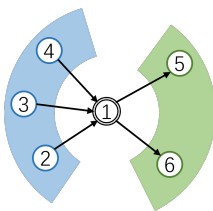

Figure 4: In directed graphs, the source nodes ($v_2, v_3, v_4$) to the central node ($v_1$) and its destination nodes ($v_5, v_6$) construct different neighbourhoods.

In the context of directed graphs, we adhere to the findings of Rossi et al. (2023), which suggest that leveraging the directionality of edges can lead to substantial improvements, particularly in heterophilic graphs. To adapt NT for directed graphs, we introduce a directed variant called Dir-NT. This variant differentiates between the source neighbours (nodes that have edges pointing towards the central node) and the destination neighbours (nodes that the central node points to), as shown in Figure 4. We apply two separate NT instances to these distinct sets of neighbours. The mathematical expression for this approach is as follows:

$$\text{NT}_1(\boldsymbol{X}, \boldsymbol{E}) + \text{NT}_2(\boldsymbol{X}, \boldsymbol{E}'), \text{ where } \boldsymbol{E}'_{(k,j),:} = \boldsymbol{E}_{(j,k),:}, \forall e_{jk} \in \mathcal{E}.$$

Here, $\text{NT}_1$ and $\text{NT}_2$ represent two separate instances of the NT. By summing the outputs of $\text{NT}_1$ and $\text{NT}_2$, we combine the information from both the source and destination neighbourhoods, allowing the model to capture the directional information in the graph and potentially improve the representation learning for nodes in directed graphs.

## 5 EXPERIMENTS

To evaluate the performance of the Neighbourhood Transformer (NT) and the proposed neighbourhood partitioning strategy, we design a series of node classification experiments on a diverse set of graphs. These experiments are conducted on five heterophilic and five homophilic graphs to demonstrate the model's strong performance and the effectiveness of our partitioning strategy. The heterophilic graphs in our study are Roman Empire, A-ratings, Minesweeper, Tolokers, and Questions (Platonov et al., 2023),. The homophilic graphs include A-computer, A-photo (McAuley et al., 2015), CoauthorCS, CoauthorPhy (Shchur et al., 2018), and WikiCS (Mernyei & Cangea, 2020). An analysis on the descrepencies of neighbourhood sizes among these datasets is provided in Appendix D. For the homophilic graphs, except for WikiCS, we follow the splitting strategy from Shirzad et al. (2023), dividing the nodes into training (60%), validation (20%), and testing (20%) sets. For the rest graphs, we use the default data splits provided with the original datasets. Other detailed experimental setups are in Appendix B.

Researchers have already conducted experiments on these 10 datasets so that we can retrieve the highest accuracy scores possible for the state-of-the-art (SotA) baselines from previous works, including their original papers and leaderboards of the respective datasets (Platonov et al., 2023;

Table 1: Averaged accuracy scores and the standard deviations in 10 runs on heterophilic graphs. The best score of undirected approaches (the upper section) for each dataset is **bolded**, and the second best is underlined.

|  | Roman Empire | A-ratings | Minesweeper | Tolokers | Questions |
|---|---|---|---|---|---|
| GCN | 73.69±0.74 | 48.70±0.63 | 89.75±0.52 | 83.64±0.67 | 76.09±1.27 |
| GraphSAGE | 85.74±0.67 | 53.63±0.39 | 93.51±0.57 | 82.43±0.44 | 76.44±0.62 |
| GAT-sep | 88.75±0.41 | 52.70±0.62 | 93.91±0.35 | 83.78±0.43 | 76.79±0.71 |
| CPGNN | 63.96±0.62 | 39.79±0.77 | 52.03±5.46 | 73.36±1.01 | 65.96±1.95 |
| FSGNN | 79.92±0.56 | 52.74±0.83 | 90.08±0.70 | 82.76±0.61 | **78.86±0.92** |
| GBK-GNN | 74.57±0.47 | 45.98±0.71 | 90.85±0.58 | 81.01±0.67 | 74.47±0.86 |
| JacobiConv | 71.14±0.42 | 43.55±0.48 | 89.66±0.40 | 68.66±0.65 | 73.88±1.16 |
| GGCN | 74.46±0.54 | 43.00±0.32 | 87.54±1.22 | 77.31±1.14 | 71.10±1.57 |
| OrderedGNN | 77.68±0.39 | 47.29±0.65 | 80.58±1.08 | 75.60±1.36 | 75.09±1.00 |
| tGNN | 79.95±0.75 | 48.21±0.53 | 91.93±0.77 | 70.84±1.75 | 76.38±1.79 |
| CDE | 91.64±0.28 | 47.63±0.43 | 95.50±5.23 | — | 75.17±0.99 |
| BloomGML | 85.26±0.25 | 52.92±0.39 | 93.30±0.16 | **85.92±0.14** | 77.93±0.34 |
| **NT** | **91.71±0.57** | **54.25±0.50** | **97.42±0.50** | 85.69±0.54 | 78.46±1.10 |
| Dir-GNN | 91.23±0.32 | 47.89±0.39 | 87.05±0.69 | 81.19±1.05 | 76.13±1.24 |
| **Dir-NT** | **94.77±0.31** | **49.43±0.62** | **93.92±0.59** | **85.02±0.77** | **77.99±1.00** |

Deng et al., 2024). The baselines we compare against are a comprehensive list of GNNs, including GCN (Kipf & Welling, 2016), GraphSAGE (Hamilton et al., 2017), GAT (Velickovic et al., 2018), GAT-sep (Platonov et al., 2023), GCNII (Chen et al., 2020), GPRGNN (Chien et al., 2021), APPNP (Klicpera et al., 2019), PPRGo (Bojchevski et al., 2020), GGCN (Yan et al., 2022), OrderedGNN (Song et al., 2023), tGNN (Hua et al., 2022), CDE (Zhao et al., 2023), BloomGML (Zheng et al., 2024), FSGNN (Maurya et al., 2022), CPGNN (Zhu et al., 2021), FAGCN (Bo et al., 2021), GBK-GNN (Du et al., 2022), JacobiConv (Wang & Zhang, 2022), and Dir-GNN (Rossi et al., 2023).

## 5.1 Performance on Heterophilic and Homophilic Graphs

We showcase the capabilities of NT on five heterophilic graphs and compare its performance against SotA GNNs, which are specifically designed to handle heterophily. The average accuracy scores across 10 runs for each method on each graph are in Table 1. The table reveals that NT outperforms the existing SotA GNNs on three of the five graphs and ranks as the second best on the rest two datasets. These results provide strong evidence that NT is a robust and powerful approach for node classification tasks on heterophilic graphs, where traditional GNNs often struggle due to the lack of homophily.

In addition to the general results, we have also included the performance of Dir-GNN (Rossi et al., 2023) and our proposed Dir-NT in the lower part of Table 1. The data clearly indicate that Dir-NT surpasses Dir-GNN across all tested datasets, with a particularly impressive accuracy score of 94.77 on the Roman Empire graph. This demonstrates that Dir-NT is more effective at leveraging the directional information in edges compared to Dir-GNN. However, we observe that on the A-ratings and Minesweeper datasets, both Dir-GNN and Dir-NT underperform compared to their undirected counterparts. This discrepancy can be attributed to the nature of these datasets. Although Platonov et al. (2023) has annotated these datasets with uni-directional edges, the relationships they represent, such as co-purchased products in A-ratings and adjacent grids in Minesweeper, are inherently undirected. Consequently, modelling these graphs as directed does not provide any additional beneficial information. Instead, it splits the neighbourhood into source and destination halves, which can interfere with the full exchange of monophilic messages. Similarly, the Tolokers dataset, which represents a network of project colleagues, is also fundamentally undirected. However, its significantly higher density (more than 10 times denser than the other four graphs) means that dividing the neighbourhood into two parts has a minimal negative impact on performance.

Moreover, we showcase the adaptability of NT on five homophilic graphs and compare its performance against other SotA GNNs. The average accuracy scores from 10 independent runs for each

Table 2: Averaged accuracy scores and the standard deviations in 10 runs on homophilic graphs. The best score for each dataset is **bolded**, and the second best is underlined.

|  | A-computer | A-photo | CoauthorCS | CoauthorPhy | WikiCS |
|---|---|---|---|---|---|
| GCN | 89.65±0.52 | 92.70±0.20 | 92.92±0.12 | 96.18±0.07 | 77.47±0.85 |
| GraphSAGE | 91.20±0.29 | 94.59±0.14 | 93.91±0.13 | 96.49±0.06 | 74.77±0.95 |
| GAT | 90.78±0.13 | 93.87±0.11 | 93.61±0.14 | 96.17±0.08 | 76.91±0.82 |
| GCNII | 91.04±0.41 | 94.30±0.20 | 92.22±0.14 | 95.97±0.11 | 78.68±0.55 |
| GPRGNN | 89.32±0.29 | 94.49±0.14 | 95.13±0.09 | 96.85±0.08 | 78.12±0.23 |
| APPNP | 90.18±0.17 | 94.32±0.14 | 94.49±0.07 | 96.54±0.07 | 78.87±0.11 |
| PPRGo | 88.69±0.21 | 93.61±0.12 | 92.52±0.15 | 95.51±0.08 | 77.89±0.42 |
| GGCN | 91.81±0.20 | 94.50±0.11 | 95.25±0.05 | 97.07±0.05 | 78.44±0.53 |
| OrderedGNN | 92.03±0.13 | 95.10±0.20 | 95.00±0.10 | 97.00±0.08 | 79.01±0.68 |
| tGNN | 83.40±1.33 | 89.92±0.72 | 92.85±0.48 | 96.24±0.24 | 71.49±1.05 |
| **NT** | **92.61±0.63** | **96.12±0.39** | **96.07±0.32** | **97.32±0.11** | **80.04±0.61** |

method on each of the homophilic graphs are presented in Table 2. As the table reveals, NT achieves the highest performance scores on all five graphs. It confirms our earlier analysis that NT is capable of adapting to the homophily present in graphs, making it a robust and general framework for graph representation learning.

In summary, NT demonstrates SotA performance on a variety of graphs, regardless of whether they are heterophilic or homophilic, undirected or directed. More comparisons against SotA graph transformers can be found in Appendix C.

## 5.2 ABLATION STUDIES ON NEIGHBOURHOOD PARTITIONING

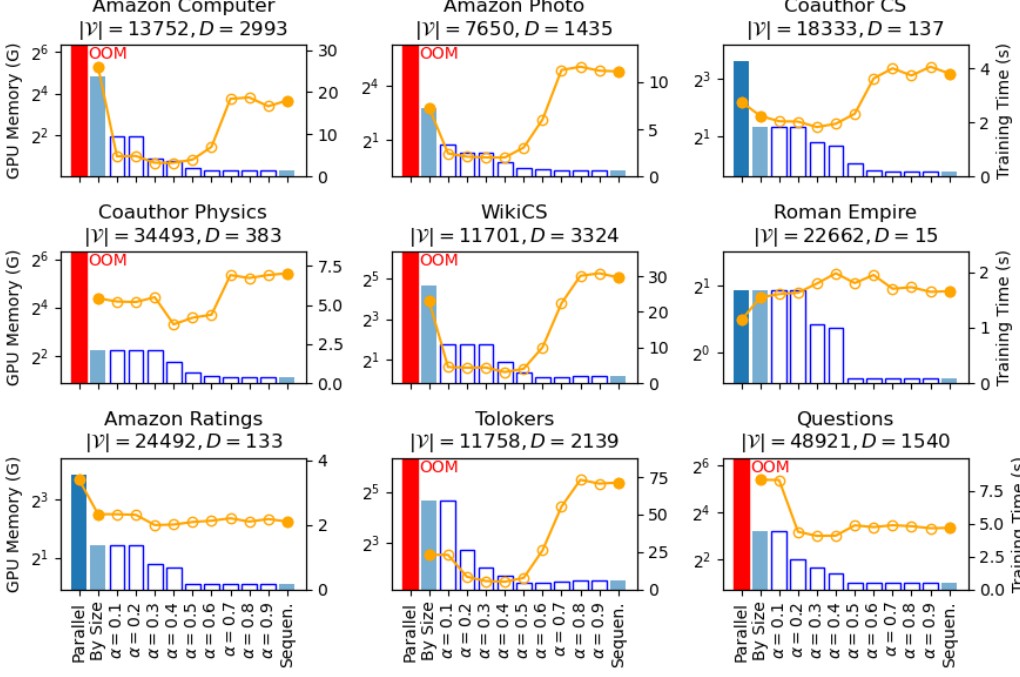

Figure 5: Ablation studies on the neighbourhood partitioning strategy. Bars represent memory footprints and curves are time consumptions. We compare paralleled processing (the first bar/point), partitioning neighbourhoods by size (the second bar/point), by both size and area with $\alpha = 0.1, 0.2, \ldots, 0.9$ (the hollow bars/points), and sequential processing (the last bar/point).

In this section, we benchmark NT using nine datasets to elucidate the attributes of the neighbourhood partitioning strategy. Figure 5 provides a visual representation of the GPU memory consumption and training time for parallel processing, partitioning neighbourhoods by size, by both size and area, and sequential processing. Parallel processing runs out of 80GB memory (OOM, depicted as red bars) in six out of the nine datasets and is only feasible on graphs with a low maximum node degree $D$. By partitioning neighbourhoods by size and incorporating Performer, the memory footprint is dramatically decreased to below 30GB across all graph datasets. With additional partitionings by area, the memory footprint is further reduced, for instance, to less than 4GB when $\alpha = 0.4$. This reduction in GPU memory utilization is crucial for the practical application of NT.

Moreover, as illustrated in the figure, the training time curves exhibit a bowl-shaped pattern with the minimum points occurring around $\alpha = 0.4$, showing that our method can be an order of magnitude faster than sequential processing (e.g., 8.63 times faster on WikiCS and 12.64 times faster on Tolokers). The only exception is observed on the Roman Empire graph, where our approach is 16% slower compared to sequential processing. This discrepancy arises due to the highly concentrated distribution of node degrees in Roman Empire, which has an average node degree of 2.91. In such a scenario, the simplicity of the graph structure allows sequential processing to handle all neighbourhoods within a limited number of operations, negating the benefits of our partitioning strategy.

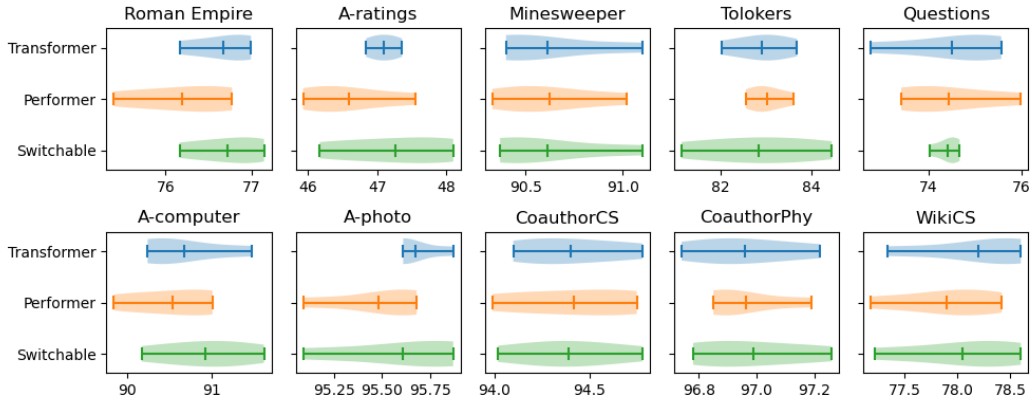

Figure 6: Testing accuracy scores (%, the horizontal axes) in 10 runs of different attention modules: Transformer with self-attention, Performer with linear-attention, and our switchable attention module. The vertical line in the centre of each violin plot represents the average score.

As the above experiments indicate, the integration of our switchable attention module with the linear attention mechanism of Performer substantially diminishes the memory and computational demands of NT. To investigate whether this integration impairs node classification accuracy, we conducted an ablation study focusing on the attention module. Figure 6 displays the accuracy scores for different attention modules across ten datasets. As depicted in the figure, while the Performer with linear attention lags on Roman Empire, A-ratings, A-computer, A-photo, and WikiCS, there is no statistically significant discrepancy in accuracy between the Transformer with full-rank self-attention and our proposed switchable attention module. Consequently, we deduce that switching between different attentions does not degrade classification accuracy.

In summary, our neighbourhood partitioning strategy successfully diminishes the spatial and temporal complexities associated with NT, all while maintaining its performance integrity.

## 6 CONCLUSIONS

We introduce Neighbourhood Transformers (NT) designed to exploit the universal monophily observed in real-world graphs. This exploitation allows NT to effectively address heterophily and to be adaptive in homophilic graphs. We overcome the space and computational challenges inherent to NT with a neighbourhood partitioning strategy, thereby enabling the practical implementation of NT on standard hardware.

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

# A ABLATION STUDIES

## A.1 ON AGGREGATORS

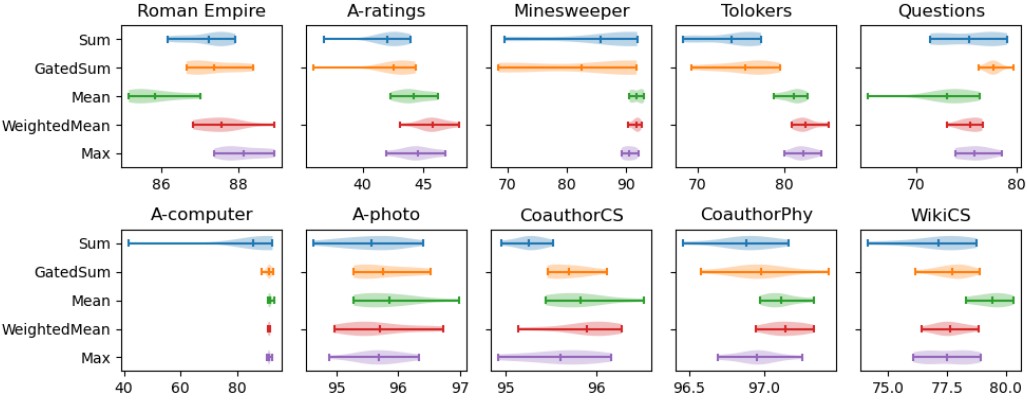

Figure 7: Testing accuracy scores (%, the horizontal axes) in 10 runs of different aggregators. The vertical line in the centre of each violin plot represents the average score.

In this section, we perform an ablation study on the aggregator used within our Neighbourhood Transformers (NT). We evaluate five different aggregators: mean, weighted-mean, sum, gated-sum, and max. As shown in Figure 7, there is no clear trend in performance across different aggregators, with the exception that the sum aggregator tends to be unstable and often results in worse performance. Among the tested aggregators, weighted-mean appears to be a more robust choice overall. However, the gated-sum aggregator achieves the highest score on the Questions dataset, while the mean aggregator performs best on the WikiCS dataset. This suggests that the choice of aggregator can significantly impact the performance of NT and that the optimal aggregator may vary depending on the specific characteristics of the dataset.

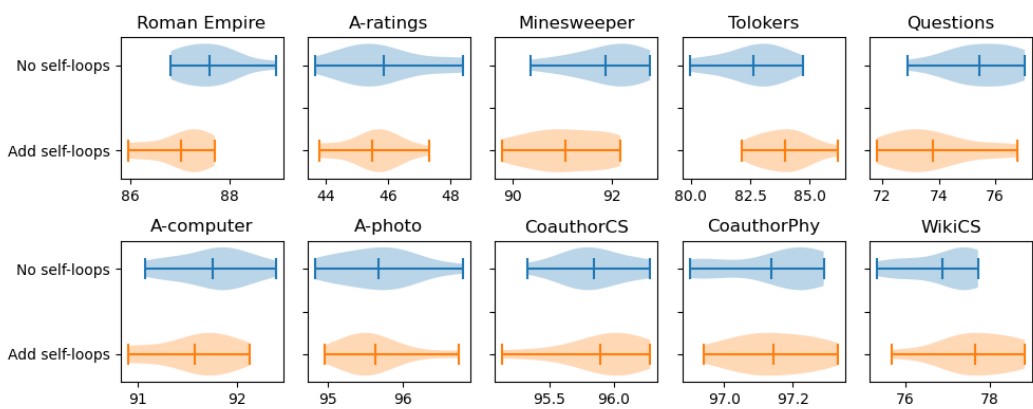

Figure 8: Testing accuracy scores (%, the horizontal axes) in 10 runs of whether to add self-loops. The vertical line in the centre of each violin plot represents the average score.

## A.2 ON SELF-LOOPS

In this section, we investigate the impact of adding self-loops to nodes in NT. Adding self-loops, which are edges connecting a node to itself, is a technique often used to modify the graph spectrum and facilitate the learning of smoother representations (Wu et al., 2019). When self-loops are added in NT, each node effectively becomes part of its own neighbourhood. This inclusion introduces an inductive bias towards homophily, as it assumes that nodes are similar to their neighbours, which may not always be the case in heterophilic graphs. The ablation study presented in this section, as depicted in Figure 8, reveals that incorporating self-loops can occasionally degrade performance on heterophilic graphs due to the aforementioned incorrect inductive bias. In contrast, for homophilic graphs, the addition of self-loops does not lead to an accuracy increase on 4 out of 5 datasets. This suggests that NT is already adept at capturing the homophily present in these graphs, and thus, the extra self-loops do not contribute additional benefits. These findings highlight the importance of considering the underlying graph structure and the nature of the relationships between nodes when deciding on the use of self-loops in graph representation learning models like NT.

## A.3 ON EMBEDDINGS SEPARATION

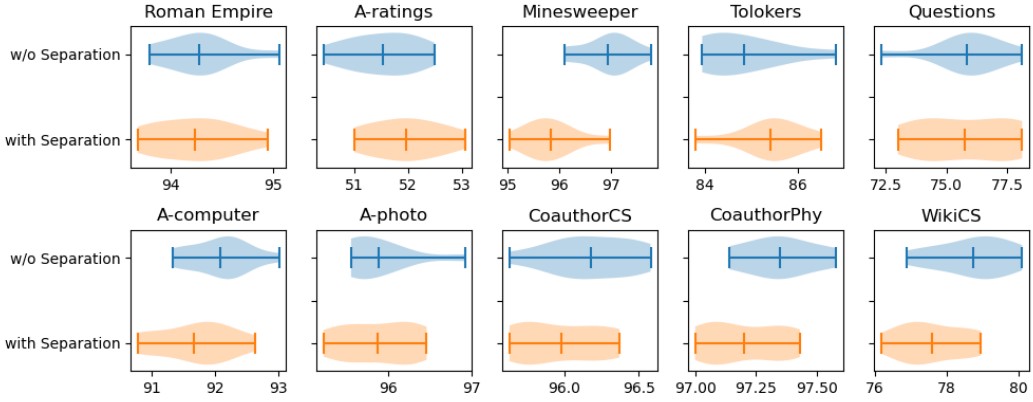

Figure 9: Testing accuracy scores (%, the horizontal axes) in 10 runs of whether to separate ego- and neighbour-embeddings. The vertical line in the centre of each violin plot represents the average score.

Table 3: Statistics of datasets in our experiments

| | Homophily (%) | #Nodes | #Edges | Mean Deg. | Features | Classes |
|---|---|---|---|---|---|---|
| Roman Empire | -4.68 | 22662 | 32927 | 2.91 | 300 | 18 |
| A-ratings | 14.02 | 24492 | 93050 | 7.60 | 300 | 5 |
| Minesweeper | 0.94 | 10000 | 39402 | 7.88 | 7 | 2 |
| Tolokers | 9.26 | 11758 | 519000 | 88.28 | 10 | 2 |
| Questions | 2.07 | 48921 | 153540 | 6.28 | 301 | 2 |
| A-computer | 68.23 | 13752 | 245861 | 35.76 | 767 | 10 |
| A-photo | 78.50 | 7650 | 119081 | 31.13 | 745 | 8 |
| CoauthorCS | 78.45 | 18333 | 81894 | 8.93 | 6805 | 15 |
| CoauthorPhy | 87.24 | 34493 | 247962 | 14.38 | 8415 | 5 |
| WikiCS | 57.90 | 11701 | 216123 | 36.85 | 300 | 10 |

Zhu et al. (2020) shows that the challenge posed by heterophily in graphs can be mitigated by distinguishing between ego-embeddings (representations of the central node itself) and neighbour-embeddings (representations of the node's neighbours) during the aggregation process. Can this experience of message passing (MP) be brought into NT? We answer this question by presenting Figure 9, which shows the outcomes of our study on whether to implement this separation in NT. The results indicate that except for the A-ratings and Tolokers datasets, adopting the separation of ego- and neighbour-embeddings leads to a decline in performance on 8 out of 10 graphs. The rationale behind this is that in heterophilic graphs, the central node tends to be dissimilar to its neighbouring nodes. Therefore, these dissimilar nodes require different transformations before aggregating and the separation trick addresses this requirement. However, NT is to aggregate messages from 2-hop neighbours, which are similar to the targeted node according to monophily. Thus, this separation is not necessary. This finding suggests that NT's inherent ability to handle the aggregation of node information may already be sufficient to capture the complex relationships in heterophilic graphs, rendering the separation of ego- and neighbour-embeddings an unnecessary step for improving performance in most cases.

## B  EXPERIMENTAL DETAILS

Table 3 provides a detailed description of the 10 datasets utilized in our experimental analysis. The first five datasets are classified as heterophilic graphs (Platonov et al., 2023). The latter five datasets are identified as homophilic graphs (Shchur et al., 2018; Mernyei & Cangea, 2020). To quantify the degree of homophily within a graph, we use the adjusted homophily metric, as introduced in Platonov et al. (2023). It is evident from the measurements that heterophilic graphs exhibit lower homophily scores across the board.

Our experimental setup involves the integration of NT into the GAT-sep architecture, as proposed in Platonov et al. (2023). Specifically, the network architecture is structured as follows: it begins with a linear encoder, followed by $L$ residual blocks, and concludes with a linear predictor. Each residual block incorporates a skip connection (He et al., 2015) and consists of a layer normalization layer, an NT layer, and a two-layer multi-layer perceptron (MLP). For model training, we utilize the Adam optimizer (Kingma & Ba, 2015).

When conducting experiments that produce results of NT in Table 1 and Table 2, the training process is limited to a maximum of 2500 epochs and employs an early stopping strategy to halt training if the performance on the validation set stagnates for 500 consecutive epochs. The learning rate for the optimizer is 0.001. Other hyperparameters are tuned using We use Optuna (Akiba et al., 2019) to search aggregator in mean, weighted-mean, sum, gated-sum, and max, the number of hidden dimensions in each attention head from 8 to 64, the number of attention heads from 1 to 8, the number of NT layers from 1 to 5, and dropout in $\{0.1, 0.2, \ldots, 0.8\}$. The optimal hyperparameters we found are summarized in Table 4. The same hyperparameter space is also searched to get the results in Figure 9. Other ablation studies are manually assigned with experimental settings as described in Table 5.

Table 4: Hyperparameters of NT on 10 graphs.

|  | Aggregator | #dimensions | #heads | #layers | Dropout |
|---|---|---|---|---|---|
| Roman Empire | sum | 32 | 6 | 5 | 0.4 |
| A-ratings | mean | 40 | 8 | 1 | 0.3 |
| Minesweeper | sum | 53 | 1 | 5 | 0.2 |
| Tolokers | gated-sum | 30 | 2 | 5 | 0.1 |
| Questions | sum | 32 | 4 | 1 | 0.2 |
| Roman Empire (directed) | max | 36 | 5 | 5 | 0.4 |
| A-ratings (directed) | max | 23 | 7 | 4 | 0.4 |
| Minesweeper (directed) | sum | 15 | 2 | 5 | 0.1 |
| Tolokers (directed) | gated-sum | 9 | 4 | 4 | 0.2 |
| Questions (directed) | gated-sum | 27 | 7 | 1 | 0.3 |
| A-computer | sum | 17 | 4 | 5 | 0.4 |
| A-photo | mean | 18 | 7 | 4 | 0.6 |
| CoauthorCS | weighted-mean | 41 | 8 | 2 | 0.3 |
| CoauthorPhy | weighted-mean | 16 | 2 | 2 | 0.1 |
| WikiCS | mean | 38 | 1 | 3 | 0.2 |

Table 5: Experimental settings of NT on ablation studies.

|  | Figure 5 | Figure 6 | Figure 7 | Figure 8 |
|---|---|---|---|---|
| Aggregator | mean | weighted-mean | — | weighted-mean |
| #dimensions | 8 | 8 | 8 | 8 |
| #heads | 4 | 4 | 8 | 8 |
| #layers | 1 | 1 | 2 | 2 |
| Dropout | 0 | 0.2 | 0.2 | 0.2 |
| Learning rate | 0.01 | 0.01 | 0.01 | 0.01 |
| #Epochs | 500 | 1000 | 200 | 200 |
| Early stop | 50 | 200 | 200 | 200 |

For the experiments that yield the results of NT presented in Table 1 and Table 2, the training protocol is constrained to a maximum of 2500 epochs. An early stopping mechanism is implemented to terminate training when there is no improvement in the validation set performance for 500 consecutive epochs. The learning rate for the optimizer is set at $0.001$. The selection of other hyperparameters is facilitated by Optuna (Akiba et al., 2019), which is used to perform a search over the following parameters: the aggregator type, including mean, weighted-mean, sum, gated-sum, and max; the number of hidden dimensions per attention head, ranging from 8 to 64; the number of attention heads, ranging from 1 to 8; the number of layers, ranging from 1 to 5; and the dropout rate, which is searched within the set $\{0.1, 0.2, \ldots, 0.8\}$. The optimal hyperparameters identified through this search are summarized in Table 4. The identical hyperparameter space is also explored to obtain the results presented in Figure 9. Additional ablation studies are conducted with manually assigned experimental settings, as detailed in Table 5.

## C  COMPARING WITH GRAPH TRANSFORMERS

We illustrate the comparison between NT and state-of-the-art graph transformers (GT) with Table 6 and Table 7 by referencing the latest data from Polynormer's publication (Deng et al., 2024), which also includes GraphGPS (Rampásek et al., 2022), NAGphormer (Chen et al., 2023), Exphormer (Shirzad et al., 2023), NodeFormer (Wu et al., 2022), DIFFormer (Wu et al., 2023), and GOAT (Kong et al., 2023).

In Table 6, we observe that NT outperforms GT on the Roman Empire dataset by a significant margin due to its utilization of directional information. On the contrary, GT layers do not consider the edges of nodes, thus they are unable to model the directionality of edges to achieve optimal performance on such graphs. On other heterophilic datasets, NT exceeds most GTs and is only slightly behind

Table 6: Averaged accuracy scores and the standard deviations in 10 runs on heterophilic graphs. The best score of undirected approaches (the upper section) for each dataset is **bolded**, and the second best is underlined.

| | Roman Empire | A-ratings | Minesweeper | Tolokers | Questions |
|---|---|---|---|---|---|
| GraphGPS | 82.00±0.61 | 53.10±0.42 | 90.63±0.67 | 83.71±0.48 | 71.73±1.47 |
| NAGphormer | 74.34±0.77 | 51.26±0.72 | 84.19±0.66 | 78.32±0.95 | 68.17±1.53 |
| Exphormer | 89.03±0.37 | 53.51±0.46 | 90.74±0.53 | 83.77±0.78 | 73.94±1.06 |
| NodeFormer | 64.49±0.73 | 43.86±0.35 | 86.71±0.88 | 78.10±1.03 | 74.27±1.46 |
| DIFFormer | 79.10±0.32 | 47.84±0.65 | 90.89±0.58 | 83.57±0.68 | 72.15±1.31 |
| GOAT | 71.59±1.25 | 44.61±0.50 | 81.09±1.02 | 83.11±1.04 | 75.76±1.66 |
| Polynormer | 92.55±0.37 | **54.81±0.49** | **97.46±0.36** | **85.91±0.74** | **78.92±0.89** |
| **NT** | **94.77±0.31** | 54.25±0.50 | 97.42±0.50 | 85.69±0.54 | 78.46±1.10 |

Table 7: Averaged accuracy scores and the standard deviations in 10 runs on homophilic graphs. The best score for each dataset is **bolded**, and the second best is underlined.

| | A-computer | A-photo | CoauthorCS | CoauthorPhy | WikiCS |
|---|---|---|---|---|---|
| GraphGPS | 91.19±0.54 | 95.06±0.13 | 93.93±0.12 | 97.12±0.19 | 78.66±0.49 |
| NAGphormer | 91.22±0.14 | 95.49±0.11 | 95.75±0.09 | **97.34±0.03** | 77.16±0.72 |
| Exphormer | 91.47±0.17 | 95.35±0.22 | 94.93±0.01 | 96.89±0.09 | 78.54±0.49 |
| NodeFormer | 86.98±0.62 | 93.46±0.35 | 95.64±0.22 | 96.45±0.28 | 74.73±0.94 |
| DIFFormer | 91.99±0.76 | 95.10±0.47 | 94.78±0.20 | 96.60±0.18 | 73.46±0.56 |
| GOAT | 90.96±0.90 | 92.96±1.48 | 94.21±0.38 | 96.24±0.24 | 77.00±0.77 |
| Polynormer | **93.68±0.21** | **96.46±0.26** | 95.53±0.16 | 97.27±0.08 | **80.10±0.67** |
| **NT** | 92.61±0.63 | 96.12±0.39 | **96.07±0.32** | 97.32±0.11 | 80.04±0.61 |

**Polynormer.** We would like to emphasize that the hyperparameter budgets for Polynormer are higher than those for NT. Specifically, even without considering the additional GT layers, the maximum number of its convolutional layers is 10, while for NT, it is 5. This is because many of our (and also Performer's) baselines in Table 1 and Table 2, which are cited from Platonov et al. (2023), are with this low-budget settings. In Table 7, NT is ranked as one of the top 2 methods across all five homophilic datasets, with the best average rank of $(2 + 2 + 1 + 2 + 2)/5 = 1.8$, which is better than Polynormer's average rank of $(1 + 1 + 4 + 3 + 1)/5 = 2$.

In conclusion, NT demonstrates competitive performance against the SoTA GTs across both heterophilic and homophilic datasets.

However, we would like to emphasize additionally that, as analyzed in the Related Works, GTs are unable to replace Message Passing Neural Networks (MPNN) as an independent method due to the information loss of topology. Consequently, the current GT methods are usually combined with MPNNs (e.g. GraphGPS) and would be more accurately described as 'GT-augmented MPNNs'. In contrast, our research demonstrates that NT is capable of handling heterophily and can be adaptively compatible with MP, potentially replacing it as an alternative component in future GNN architectures. This is actually orthogonal to the GT approach; that is, NT can also be combined with GT to form an enhanced 'GT-augmented NT'.

## D  ANALYSIS ON THE DESCREPENCIES OF NEIGHBOURHOOD SIZES IN TRAINING AND IN INFERENCE

To check if NT performs consistently when neighbourhood sizes are shifted from training to the inference stage, we conduct an analysis to measure the discrepancy between neighbourhood sizes in the training set and beyond.

We first approximate the averaged size of belonging neighbourhoods for each node using $s = \deg(A^2)/\deg(A)$, where $A$ is the adjacency matrix and $\deg(\cdot)$ is a function to derive node degrees

Table 8: Descrepencies of Neighbourhood sizes in training and in inference for the 10 graphs.

| Roman Empire | A-ratings | Minesweeper | Tolokers | Questions |
|---|---|---|---|---|
| 3.5% | 5.3% | 0.6% | 8.8% | 3.9% |
| A-computer | A-photo | CoauthorCS | CoauthorPhy | WikiCS |
| 6.5% | 8.8% | 6.5% | 7.2% | 14.4% |

from the adjacency matrix. Then, with 100 histogram bins, we transform elements of $s$ corresponding to the training set to distribution $P$ and the other elements to distribution $Q$. After that, we calculate the discrepency between the two distributions as $\sum_i |p_i - q_i|$, where $p_i$ is the probability of the $i$-th histogram bin of $P$ and $q_i$ is that of $Q$.

The discrepencies for the 10 datasets are reported in Table 8, indicating varying levels of discrepancy across the datasets, with Minesweeper showing low discrepancy and WikiCS showing high discrepancy. Despite these variations, NT maintains consistent performance, as demonstrated by the experiments in Table 1 and Table 2.

# E  THEORETICAL ANALYSIS ON NEIGHBOURHOOD TRANSFORMERS

Here, We outline some theoretical foundations that underpin our approach.

**Theorem 1.** *When the combiner concentrates on information from central nodes of neighbourhoods, the Neighbourhood Transformer is a message passing layer.*

*Proof.* When Equation 1 omits $\boldsymbol{H}'_{k,:}$ and becomes

$$\boldsymbol{Z}_{(j,k),:} = \phi(\text{Combiner}(\boldsymbol{H}'_{j,:})) \triangleq \boldsymbol{Z}^{(j)},$$

Equation 2 becomes a simple transformation of $\boldsymbol{Z}^{(j)}$ as

$$\boldsymbol{M}^{(j)} = \phi(\text{SelfAttention}(\oplus\{\boldsymbol{Z}^{(j)}, \boldsymbol{Z}^{(j)}, \ldots, \boldsymbol{Z}^{(j)}\})) = \begin{bmatrix} \phi(\boldsymbol{Z}^{(j)}\boldsymbol{W}_v) \\ \phi(\boldsymbol{Z}^{(j)}\boldsymbol{W}_v) \\ \cdots \\ \phi(\boldsymbol{Z}^{(j)}\boldsymbol{W}_v) \end{bmatrix}.$$

Then, the output of Equation 3 is actually equivalent to the output of a message passing layer:

$$\boldsymbol{H}_{i,:} = \text{Aggregator}(\boldsymbol{M}^{(j)}_{(i),:}|v_j \in \mathcal{N}(v_i)) = \text{Aggregator}(\phi(\phi(\text{Combiner}(\boldsymbol{H}'_{j,:})) \cdot \boldsymbol{W}_v)|v_j \in \mathcal{N}(v_i)).$$

$\square$

Theorem 1 demonstrates that NT is compatible with message passing (MP) and possesses superior or at least equivalent expressiveness. This ensures that NT can leverage the proven strengths of MP when dealing with homophily, while still potentially offering additional benefits.

**Theorem 2.** *When the combiner omits information from central nodes of neighbourhoods, the Neighbourhood Transformer with linear-attention is a two-layered message passing network.*

*Proof.* When omitting $\boldsymbol{H}'_{j,:}$ in Equation 1 and using linear-attention in Equation 2, the final representations of Equation 3 are

$$\boldsymbol{H}_{i,:} = \text{Aggregator}(\{\phi(\frac{\hat{\boldsymbol{Q}}_{i,:} \cdot \boldsymbol{K}^{(j)}_v}{\hat{\boldsymbol{Q}}_{i,:} \cdot \boldsymbol{K}^{(j)}_1})|v_j \in \mathcal{N}(v_i)\}),$$

where $\boldsymbol{K}^{(j)}_v$ and $\boldsymbol{K}^{(j)}_1$ indicates the formulas $\hat{\boldsymbol{K}}^T \cdot \boldsymbol{V}$ and $\hat{\boldsymbol{K}}^T \cdot \boldsymbol{1}_{n \times 1}$ of Performer applied in the neighbourhood $\mathcal{N}(v_j)$. In detail, the element at the position $(x, y)$ of $\boldsymbol{K}^{(j)}_v$ is $\sum_{v_k \in \mathcal{N}(v_j)} \hat{K}_{k,x} V_{k,y}$ and the $x$-th element of $\boldsymbol{K}^{(j)}_1$ is $\sum_{v_k \in \mathcal{N}(v_j)} \hat{K}_{k,x}$.

Regarding $\boldsymbol{K}_v^{(j)}$ and $\boldsymbol{K}_1^{(j)}$ as the result of a message passing layer, which aggregates information of $\mathcal{N}(v_j)$ to node $v_j$, NT can be rewritten as a two-layered message passing network, as

$$\boldsymbol{Z}^{(1)} = (\hat{\boldsymbol{K}}, \boldsymbol{V}) = (\exp(\boldsymbol{X}\boldsymbol{W}_k\boldsymbol{P} - \frac{||\boldsymbol{X}\boldsymbol{W}_k||^2}{2}), \boldsymbol{X}\boldsymbol{W}_v),$$

$$\boldsymbol{H}_{i,:}^{(1)} = (\boldsymbol{K}_v^{(i)}, \boldsymbol{K}_1^{(i)}) = ([\sum_{v_j \in \mathcal{N}(v_i)} \hat{K}_{j,x} \cdot V_{j,y}]_{xy}, [\sum_{v_j \in \mathcal{N}(v_i)} \hat{K}_{j,x}]_x),$$

$$\boldsymbol{Z}^{(2)} = \boldsymbol{H}^{(1)},$$

$$\boldsymbol{H}_{i,:}^{(2)} = \text{Aggregator}'(\exp(\frac{1}{\sqrt{h}} \cdot \boldsymbol{X}_{i,:}\boldsymbol{W}_q\boldsymbol{P}), \{\boldsymbol{Z}_{j,:}^{(2)} | v_j \in \mathcal{N}(v_i)\}),$$

where $\boldsymbol{X}$ is the inputted node features. $\qquad\square$

Theorem 2 implies that, with simplifications, NT is a message passing layer that utilizes information from 2-hop neighbours. This can be beneficial for capturing the monophilic patterns when handling heterophily.

# F  THEORETICAL ANALYSIS ON THE PARTITIONING ALGORITHM

To formalize our analysis on Algorithm 1, we outline the following two assumptions to measure the memory usage in neighbourhood processing.

**Assumption 1** (Paralleled Processing). *The memory consumption of applying the transformer to a group of neighbourhoods is proportional to the group's area, defined as the product of the number of neighbourhoods and their maximum size.*

**Assumption 2** (Sequential Processing). *Sequentially processing two neighbourhood groups with areas $s_1$ and $s_2$ consumes memory proportional to $\max(s_1, s_2)$.*

With these assumptions in place, we are able to state and prove the following theorem:

**Theorem 3.** *Given any partitioning that divides neighbourhoods into two groups, there exists an alternative partitioning that requires the same or less processing memory, where all neighbourhoods in one group are not smaller than those in the other group.*

*Proof.* Considering a partitioning where group $G_1$ contains $c_1$ neighbourhoods and group $G_2$ contains $c_2$ neighbourhoods, with the maximum size $d_1$ of neighbourhoods in $G_1$ being not smaller than that $d_2$ of $G_2$ ($d_1 \geq d_2$). If the smallest neighbourhood (with size $d_3$) in $G_1$ is smaller than the largest one (with size $d_2 > d_3$) in $G_2$, we can swap their positions to create two new groups $G_1'$ and $G_2'$. The area of $G_1'$ remains unchanged since $\max(d_1, d_2) \times c_1 = d_1 \times c_1$, while the area of $G_2'$ is unchanged or reduced since $\min(d_3, d_4) \times c_2 \leq d_2 \times c_2$, where $d_4$ is the size of the second-largest neighbourhood in $G_2$. We can keep swapping the smallest neighbourhood in the first group with the largest neighbourhood in the second group if the former is smaller than the latter until any neighbourhood in the first group is not smaller than those in the second, with the processing memory remaining unchanged or reduced.

$\qquad\square$

From Theorem 3, we conclude that the optimal partitioning can be achieved by first ordering the neighbourhoods and then scanning linearly for the partitioning point, as done in line 8 of Algorithm 1. The complexity of this search is $O(n \log n)$, accounting for the sorting step.

