# OpenReview forum: "Monophilic Neighbourhood Transformers"
_ICLR.cc/2025/Conference — Submitted to ICLR 2025_

### Official Review · Reviewer_zjDW · 2024-11-01

**Soundness:** 2
**Presentation:** 3
**Contribution:** 2
**Rating:** 5
**Confidence:** 4

**Summary:**

This paper studies the node classification problem on graphs beyond homophily. Its proposed technique is very straightforward, a variant of graph Transformer whose messages will have an extra interaction before aggregating into the node embeddings.

**Strengths:**

S1. This paper's presentation is good and clear. I can easily follow most of the content.

S2. The experimental section should be comprehensive and cover 10 datasets, including both homophilic and heterophilic graphs.

S3. From Table 1 and Table 2, the performance improvement against the SOTA is decent and appreciated.

**Weaknesses:**

W1. This paper claims that the proposed method has good expressiveness. However, I found no (theoretical) analysis regarding the expressiveness.

W2. The proposed method is actually pretty simple, and the rationale is simple, monophyly (lines 50-52), i.e., 2-hop neighbors are helpful for node classification on both homophilic and heterophilic graphs. Such a simplicity is good, but I found some defects unsolved.

W2.1 Why are 2-hop neighbors useful for node classification on graphs regardless of whether they are homophilic or heterophilic?

W2.2 It looks like the proposed method (from Eq. 1 to 3, Figure 2 c) still utilizes information from 1-hop neighbors. In my understanding, the core idea of this paper is to collect edge messages (from 1-hop neighbors) by Eq. 3 and then let the edge messages interact via a self-attention module Eq. 2. However, overall, no information from 2-hop neighbors is included. Again, this method is simple, but it is highly unclear why it is effective.

W2.3 Why previous methods cannot capture the information from 2-hop neighbors? If the 2-hop information is that useful, I think many previously proposed methods should be able to capture it. E.g., GPRGNN [1], ACM-GNN [2].


*Minor weaknesses: *

W3. Some baselines are missing [3-5]. After adding them back, the performance advantage of the proposed method is not that significant on datasets Roman Empire, A-ratings, and Tolokers.

W4 The strategies proposed in Section 4.2 are a bit heuristic-based.

W5. The code is not provided, which lowers the reproducibility of this study.

[1] Chien, Eli, et al. "Adaptive Universal Generalized PageRank Graph Neural Network." International Conference on Learning Representations.

[2] Luan, Sitao, et al. "Revisiting heterophily for graph neural networks." Advances in neural information processing systems 35 (2022): 1362-1375.

[3] Zhao, Kai, et al. "Graph neural convection-diffusion with heterophily." Proceedings of the Thirty-Second International Joint Conference on Artificial Intelligence. 2023.

[4] Jang, Hyosoon, et al. "Diffusion probabilistic models for structured node classification." Advances in Neural Information Processing Systems 36 (2024).

[5] Zheng, Amber Yijia, et al. "Graph Machine Learning through the Lens of Bilevel Optimization." International Conference on Artificial Intelligence and Statistics. PMLR, 2024.

**Questions:**

Please check the weaknesses I mentioned.

---

> ### Author Response · Authors · 2024-11-15
> **W2: The simplicity of NT is good, but I found some defects unsolved.**
>
> Dear Reviewer,
>
> We are grateful for your insightful comments and the opportunity to clarify our work.
> We have reevaluated your concerns and would like to address them in a sequence that we hope will provide a clearer understanding of our partitioning algorithm (**W2**), before proceeding to other matters such as baselines (**W3**) and theoretical analysis (**W1 & W4**).
>
> We acknowledge that our initial explanation may not have fully conveyed the intricacies of our method.
> To rectify this, we would like to walk you through our approach, using Eq.1 to Eq.3 and corresponding to Figure 2c, with a focus on calculating the representation of node 1 (the double-circled node in Figure 2c).
>
> **Firstly**, your interpretation of Eq.3 in **W2.2** is correct.
> This step involves collecting edge messages from 1-hop neighbours.
> In Figure 2c, the information of node 1 and node 2 is mixed and encoded as a blue square.
> **Subsequently**, Eq.2 represents the self-attention mechanism where all edge messages (squares) engage in message exchanging within their belonging neighborhoods.
> This is depicted in Figure 2c as the interactions among squares of the same colors.
> **Finally**, Eq.1, the aggregator, combines the edge messages related to node 1 (the three squares surrounding node 1 in the figure) to form node 1's representation.
> It is important to note that these aggregated edge messages have already undergone message exchanging within their neighborhoods, thus incorporating 2-hop information (**W2.2**).
> For instance, the blue square aggregated by node 1 also includes information from nodes 3, 4, 5, and 6.
>
> With this understanding, we address your points **W2.1**, **W2.2**, and **W2.3**.
>
> ## W2.1 Why are 2-hop neighbors useful for node classification on graphs regardless of whether they are homophilic or heterophilic?
>
> We appreciate your grasp of how NT handles heterophily through the monophily property of 2-hop neighbors.
> Regarding homophily, NT's effectiveness does not rely on the usefulness of 2-hop information but on its adaptability of switching to use 1-hop information.
> When a graph is homophilic and benefits from 1-hop information processing, NT can effectively reduce its complexity to operate like a normal GNN based on message passing, as mentioned in lines 104 to 106.
> This occurs when NT in Eq.3 primarily uses the information from the neighbours, reduces self-attention of Eq.2 to a simple feed-forward transformation.
> In essence, NT can flexibly utilize information from both 1-hop and 2-hop neighbors, making it suitable for both homophilic and heterophilic graphs.
>
> ## W2.2 It looks like the proposed method still utilizes information from 1-hop neighbors. No information from 2-hop neighbors is included.
>
> As previously explained, a node's representation is derived from all its 'belonging' neighborhoods, a.k.a. neighbours' neighbourhoods, which inherently includes information from neighbors of neighbors.
>
> ## W2.3 Why previous methods cannot capture the information from 2-hop neighbors? If the 2-hop information is that useful, I think many previously proposed methods should be able to capture it.
>
> You are correct that several previous methods have incorporated 2-hop information to address heterophily, as discussed in the Introduction and Related Works sections.
> However, these methods often rely on the second-order adjacency matrix, which can result in the loss of certain information, such as the specific path through which a 2-hop neighbor is connected.
> Our method, as outlined in Eq.3, preserves this information, ensuring that message exchanging within each neighborhood is uniquely conditioned on the central node.
> This enhances the diversity of exchanging patterns and the expressiveness of the aggregated representations, as detailed in lines 101 to 104.
>
> We trust that this detailed explanation clarifies our NT method and its novel aspects.

---

> > ### Author Response · Authors · 2024-11-15
> > **W3: Some baselines are missing.**
> >
> > We are immensely grateful for your valuable suggestions.
> > We have procured the data for the methods you mentioned: CDE, DPM-SNC, and BloomGML, and have incorporated them into the table as shown below.
> > Please note that this table does not include our improvement that utilizes directional information (Dir-), and "SotA in paper" refers to the best method other than NT in our manuscript.
> >
> > |                   | Roman Empire   | A-ratings      | Minesweeper    | Tolokers       | Questions      |
> > |-------------------|----------------|----------------|----------------|----------------|----------------|
> > | SotA in paper     | 88.75±0.41     | 53.63±0.39     | 93.91±0.35     | 83.78±0.43     | **78.86±0.92** |
> > | CDE               | *91.64±0.28*   | 47.63±0.43     | *95.50±5.23*   | -              | 75.17±0.99     |
> > | DPM-SNC           | 89.52±0.46     | **54.66±0.39** | -              | -              | -              |
> > | BloomGML          | 85.26±0.25     | 52.92±0.39     | 93.30±0.16     | **85.92±0.14** | 77.93±0.34     |
> > | **NT (w/o DIR-)** | **91.71±0.57** | *54.25±0.50*   | **97.42±0.50** | *85.69±0.54*   | *78.46±1.10*   |
> >
> > Upon including these additional methods, we acknowledge that NT does not consistently achieve the top ranking across all datasets.
> > However, aside from DPM-SNC, which has insufficient data to draw conclusions, we observed that CDE performs exceptionally poorly on the A-ratings and Questions datasets, and BloomGML on the Roman Empire dataset.
> > This results in their average rankings not being at the forefront: CDE has an average ranking of $(2+5+2+4)/4=3.25$, and BloomGML has an average ranking of $(5+4+4+1+3)/5=3.4$.
> > In contrast, NT maintains a general advantage with an average ranking of $(1+2+1+2+2)/5=1.6$.
> >
> > We will update our manuscript to include CDE and BloomGML, but we must note that the data for DPM-SNC is indeed too limited to be included in a comprehensive comparison.
> > We appreciate your input and thank you once again for your recommendations.

---

> ### Author Response · Authors · 2024-11-15
> **W1: This paper claims that the proposed method has good expressiveness. However, I found no (theoretical) analysis regarding the expressiveness.**
>
> Thank you for your patience and for prompting us to delve into the theoretical aspects of our work.
> We have carefully considered your suggestions and have strived to formalize the theoretical propositions to address your concerns.
> Below, we outline the theoretical foundations and propositions that underpin our approach
>
> **Proposition 1:** When the combiner concentrates on information from central nodes of neighbourhoods, the Neighbourhood Transformer is a message passing layer.
>
> As analyzed in line 223 to 225, when Eq.3 omits $\boldsymbol{H}' _{k,:}$ and $\boldsymbol{M}'^{(j)} _{(k)}$ to become
> $$\boldsymbol{Z} _{(j,k),:} = \phi(\text{Combiner}(\boldsymbol{H}' _{j,:})) \triangleq \boldsymbol{Z}^{(j)},$$
> Eq.2 becomes a simple transformation of $\boldsymbol{Z}^{(j)}$ as
> $$\boldsymbol{M}^{(j)} = \phi(\text{SelfAttention}(\oplus \\{ \boldsymbol{Z}^{(j)}, \boldsymbol{Z}^{(j)}, \ldots, \boldsymbol{Z}^{(j)} \\} ) = \begin{bmatrix} \phi(\boldsymbol{Z}^{(j)} \boldsymbol{W} _v) \\\\ \phi(\boldsymbol{Z}^{(j)} \boldsymbol{W} _v) \\\\ \cdots \\\\ \phi(\boldsymbol{Z}^{(j)} \boldsymbol{W} _v) \end{bmatrix}.$$
> Then, the final representations (Eq.1) are actually equivalent to the output of a message passing layer:
> $$\boldsymbol{H} _{i,:} = \text{Aggregator}(\\{ \boldsymbol{M}^{(j)} _{(i),:} | v _j \in \mathcal{N}(v _i) \\}) = \text{Aggregator}(\\{ \phi(\phi(\text{Combiner}(\boldsymbol{H}' _{j,:})) \cdot \boldsymbol{W} _v) | v _j \in \mathcal{N}(v _i) \\}).$$
>
> $\blacksquare$
>
> Proposition 1 demonstrates that NT is compatible with message passing (MP) and possesses superior or at least equivalent expressiveness.
> This ensures that NT can leverage the proven strengths of MP when dealing with homophily, while still potentially offering additional benefits.
>
> **Proposition 2:** When the combiner omits information from central nodes of neighbourhoods, the Neighbourhood Transformer with linear-attention is a two-layered message passing network.
>
> When omitting $\boldsymbol{H}' _{j,:}$ and $\boldsymbol{M}'^{(j)} _{(k)}$ in Eq.3 and using linear-attention in Eq., the final representations of Eq.1 are
> $$\boldsymbol{H} _{i,:} = \text{Aggregator}( \\{ \phi(\frac{\boldsymbol{\hat Q} _{i,:} \cdot \boldsymbol{K} _v^{(j)}}{\boldsymbol{\hat Q} _{i,:} \cdot \boldsymbol{K} _1^{(j)}}) | v _j \in \mathcal{N}(v _i) \\} ),$$
> where $\boldsymbol{K} _v^{(j)}$ and $\boldsymbol{K} _1^{(j)}$ indicates the formulas $\boldsymbol{\hat K}^T \cdot \boldsymbol{V}$ and $\boldsymbol{\hat K}^T \cdot \boldsymbol{1} _{n \times 1}$ of Performer applied in the neighbourhood $\mathcal{N}(v _j)$.
> In detail, the element at the position $(x, y)$ of $\boldsymbol{K} _v^{(j)}$ is $\sum\limits _{v _k \in \mathcal{N}(v _j)} \hat K _{k,x} V _{k,y}$ and the $x$-th element of $\boldsymbol{K} _1^{(j)}$ is $\sum\limits _{v _k \in \mathcal{N}(v _j)} \hat K _{k,x}$.
>
> Regarding $\boldsymbol{K} _v^{(j)}$ and $\boldsymbol{K} _1^{(j)}$ as the result of a message passing layer, which aggregates information of $\mathcal{N}(v _j)$ to node $v _j$, NT can be rewritten as a two-layered message passing network, as
> $$\begin{aligned}
> \boldsymbol{Z}^{(1)} &= (\boldsymbol{\hat K}, \boldsymbol{V}) = (\exp(\boldsymbol{X} \boldsymbol{W} _k \boldsymbol{P} - \frac{||\boldsymbol{X}\boldsymbol{W} _k||^2}{2}), \boldsymbol{X}\boldsymbol{W} _v) \\\\
> \boldsymbol{H}^{(1)} _{i,:} &= (\boldsymbol{K} _v^{(i)}, \boldsymbol{K} _1^{(i)}) = ([\sum\limits _{v _j \in \mathcal{N}(v _i)} \hat K _{j,x} \cdot V _{j,y}] _{xy}, [\sum\limits _{v _j \in \mathcal{N}(v _i)} \hat K _{j,x}] _x) \\\\
> \boldsymbol{Z}^{(2)} &= \boldsymbol{H}^{(1)} \\\\
> \boldsymbol{H}^{(2)} _{i,:} &= \text{Aggregator'}(\exp(\frac{1}{\sqrt{h}} \cdot \boldsymbol{X} _{i,:} \boldsymbol{W} _q \boldsymbol{P}), \\{ \boldsymbol{Z}^{(2)} _{j,:} | v _j \in \mathcal{N}(v _i) \\}). \\\\
> \end{aligned}$$
>
> $\blacksquare$
>
> Proposition 2 implies that, also with simplifications, NT is a message passing layer that utilizes information from 2-hop neighbours.
> This can be beneficial for capturing the monophilic patterns when handling heterophily.

---

> ### Author Response · Authors · 2024-11-15
> **W4: The strategies proposed in Section 4.2 are a bit heuristic-based.**
>
> We would like to express our gratitude for your guidance, which has been instrumental in refining the theoretical foundation of our partitioning algorithm as presented in Section 4.2.2.                                                   Below, we outline the assumptions and proposition that we have developed to formalize our analysis:
>
> **Assumption 1 on paralleled processing:** The memory consumption of applying the transformer to a group of neighbourhoods is proportional to the group's area, defined as the product of the number of neighbourhoods and their maximum size.
>
> **Assumption 2 on sequential processing:** Sequentially processing two neighbourhood groups with areas $s_1$ and $s_2$ consumes memory proportional to $\max(s_1, s_2)$.
>
> With these assumptions in place, we are able to state and prove the following proposition:
>
> **Proposition 3:** Given any partitioning that divides neighbourhoods into two groups, there exists an alternative partitioning that requires the same or less processing memory, where all neighbourhoods in one group are not smaller than those in the other group.
>
> Considering a partitioning where group $G_1$ contains $c_1$ neighbourhoods and group $G_2$ contains $c_2$ neighbourhoods, with the maximum size $d_1$ of neighbourhoods in $G_1$ being not smaller than that $d_2$ of $G_2$ ($d_1 \ge d_2$).If the smallest neighbourhood (with size $d_3$) in $G_1$ is smaller than the largest one (with size $d_2 \gt d_3$) in $G_2$, we can swap their positions to create two new groups $G'_1$ and $G'_2$.
> The area of $G'_1$ remains unchanged since $\max(d_1, d_2) \times c_1 = d_1 \times c_1$, while the area of $G'_2$ is unchanged or reduced since $\min(d_3, d_4) \times c_2 \le d_2 \times c_2$, where $d_4$ is the size of the second-largest neighbourhood in $G_2$.                                                                                                                                                                                                                   We can keep swapping the smallest neighbourhood in the first group with the largest neighbourhood in the second group if the former is smaller than the latter until any neighbourhood in the first group is not smaller than those in the second, with the processing memory remaining unchanged or reduced.
>
> $\blacksquare$
>
> From Proposition 3, we conclude that the optimal partitioning can be achieved by first ordering the neighbourhoods and then scanning linearly for the partitioning point, as done in line 8 of Algorithm 1.
> The complexity of this search is $O(n \log n)$, accounting for the sorting step.
>
> We will include the above theoretical analysis in the appendix of our paper and would like to thank you once again for highlighting the areas where our work could be strengthened.

---

> > ### Author Response · Authors · 2024-11-15
> > **W5: The code is not provided, which lowers the reproducibility of this study.**
> >
> > To address your concerns regarding the reproducibility of our study, we are pleased to provide you with the anonymous repository link containing all the experimental code necessary to reproduce Table 1 and Table 2.
> >
> > Please find the link below:
> > https://anonymous.4open.science/r/MoNT-BD3C

---

> > > ### Comment · Reviewer_zjDW · 2024-11-23
> > >
> > > I appreciate the authors' prompt response and am sorry for my slightly late response.
> > >
> > > I would like to comment on some of our discussions, and I will index them with **the number in our original review**.
> > >
> > > W2.2, thanks for the clarification. I indeed missed some details and you are right the aggregation includes 2-hop messages.
> > >
> > > W1. Thanks for your additional propositions 1 and 2. However, respectively saying, in my personal view, they are not that interesting. I think this is a shared understanding by the community that **Graph Transformers are (theoretically) more powerful than GNNs, no matter 1-hop or 2-hop ones** because the inner-product-based attention matrix can propagate messages between **any pair of nodes**, but GNNs can only propagate messages to, say, $k$-hop neighbors. Thus, your propositions claim that graph transformers can be as powerful as GNNs is somewhat like a "degradation," which is not interesting. In my expectation, if you claim your method is more expressive, there should be some theoretical expressiveness improvement such as WL expressiveness (e.g., [1]) or signal processing-based expressiveness (e.g., [2]).
> > >
> > > W2.3 Your response mentioned that "However, these methods often rely on the second-order adjacency matrix, which can result in the loss of certain information, such as the specific path through which a 2-hop neighbor is connected." This does not convince me. For example, if 2-hop neighbors are that important, GPRGNN [2] can assign a heavy weight to them. In other words, **many existing methods should be able to capture the messages from 2-hop adaptively.**
> > >
> > > W4. From the algorithmic perspective, in my personal view, this partition-based method in Section 4.2 is ad hoc and heuristic-based. I would maintain my evaluation on this technique and frankly saying, I partially agree with the W4 mentioned by the reviewer GZCm.
> > >
> > > [1] Chen, Lei, Zhengdao Chen, and Joan Bruna. "On Graph Neural Networks versus Graph-Augmented MLPs." International Conference on Learning Representations.
> > >
> > > [2] Chien, Eli, et al. "Adaptive Universal Generalized PageRank Graph Neural Network." International Conference on Learning Representations.

---

> > > > ### Author Response · Authors · 2024-11-25
> > > > **Thank you for your response. We are almost there.**
> > > >
> > > > We are deeply appreciative of your time and effort in reviewing our submission.
> > > > We would like to address the unresolved concerns **W1**, **W2.3**, and **W4** as follows.
> > > >
> > > > ## W1: There should be some theoretical expressiveness improvement such as WL expressiveness or signal processing-based expressiveness
> > > >
> > > > Upon further investigation, we concur with your perspective: although we have demonstrated that NT is not less expressive than single-layer message passing, the expressiveness of NT remains equivalent to the 1-WL test.
> > > > In the 1-WL test, if the initial colors assigned to all nodes are identical, it is tantamount to omitting information from the central nodes of neighborhoods, which negates one of NT's strengths (we will elaborate on this point in our response to W2.3 later).
> > > > As postulated in our previously proposed proposition 2, this simplification with linear-attention is akin to a two-layered MP.
> > > >
> > > > Therefore, we will retract the claims regarding expressiveness in the next version of the PDF and, depending on the context, modify the language to terms such as 'strong performance' or 'attentiveness' to avoid the previous misrepresentation, thereby solely highlighting the advantage of our method in adapting to various data distributions in graphs.
> > > > We thank you for this correction.
> > > >
> > > > Additionally, as you have directly compared Graph Transformers (GT) and NT, we would like to further emphasize that GTs are not capable of replacing Message Passing Neural Networks (MPNN) as an independent method due to the loss of topological information, as discussed in the Related Works section.
> > > > Consequently, the current GT methods are often integrated with MPNNs (e.g., GraphGPS [1]) and would be more accurately characterized as 'GT-augmented MPNNs'. In contrast, our research shows that NT is adept at handling heterophily and can be adaptively compatible with MP, potentially serving as an alternative component in future GNN architectures.
> > > > This is, in fact, orthogonal to the GT approach; that is, NT can also be integrated with GT to form an advanced 'GT-augmented NT'.
> > > > This is the primary reason we did not conduct a direct comparison between GT and NT in the main text of the manuscript. (Now we have appended the comparison to the Appendices.)
> > > >
> > > > > [1] Ladislav Rampásek, et al.: Recipe for a General, Powerful, Scalable Graph Transformer. NeurIPS 2022
> > > >
> > > > ## W2.3: Many existing methods should be able to capture the messages from 2-hop adaptively
> > > >
> > > > We hope to elucidate the distinction between NT and conventional MPNNs through the following example.
> > > > Consider a scenario where we have three types of nodes: evidence, conclusion, and agents who submit evidences to support conclusions.
> > > > We are given a graph structure represented as '{Evidence 1, Evidence 2} - Agent 1 - Conclusion - Agent 2 - {Evidence 3, Evidence 4}'.
> > > >
> > > > Using NT, we are capable of effectively aggregating the evidence to substantiate the conclusion in intricate situations where the evidence may be contradictory and the reliability of agents may vary depending on the evidences they submit.
> > > > In contrast, with conventional message-passing-based approaches such as GPRGNN, the graph is reduced to a simplified structure of '{Evidence 1, Evidence 2, Evidence 3, Evidence 4} - Conclusion' in the 2-hop perspective.
> > > > This reduction obscures critical information, such as the specific pathway (e.g., via Agents) through which a 2-hop neighbour (e.g., Evidences) is linked (e.g., to Conclusions).
> > > >
> > > > In summary, while existing methods can dynamically assign weights to 2-hop neighbours based on their own significance, NT can further provide more complex adjustments by considering their 1-hop intermediaries.

---

> > > > ### Author Response · Authors · 2024-11-25
> > > > **W4: The partition-based method is ad hoc and heuristic-based**
> > > >
> > > > We are delighted to hear your personal views, even if they are against to our partitioning method.
> > > > While we understand that the assessment of what constitutes an 'interesting' contribution may be a matter of taste and we may not be able to fully address your concerns to your satisfaction, we are committed to considering your perspective with an open mind.
> > > >
> > > > Frankly speaking, we concede that the partitioning method itself may not be as interesting.
> > > > In fact, beyond the concept of applying transformers to every neighborhood, the idea that mostly excites us in the manuscript is the switchable attention module because, to the best of our knowledge, this is the first hybrid application (with a shared set of parameters but two different algorithms) of Performer and Transformer in literatures.
> > > > Previously, the property of Performer's being fully compatibility with Transformer was primarily used in the NLP domain to transfer pre-trained Transformer knowledge to the Performer for a faster start-up and no hybrid mode is ever explored.
> > > > On the contrary, in the graph domain, the unique challenge of dealing with the uneven distribution of neighborhood sizes necessitates us to consider both efficient handling of large neighborhoods and nuanced processing of smaller ones, finally leading to the Switchable Attention.
> > > > This hybrid mode has proven to be highly successful.
> > > > On one hand, as indicated by the second bars in Figure 5, incorporating Performer significantly reduces memory consumption.
> > > > On the other hand, as shown in Figure 6, the Switchable Attention prevents the notable accuracy loss that Performer would otherwise experience on datasets such as Roman Empire, A-ratings, A-computer, A-photo, and WikiCS.
> > > >
> > > > Regarding the partitioning method, it plays a **fundamental** role in supporting the two most interesting components: NT and Switchable Attention.
> > > > Its additional merits include being **fast** (by reducing the complexity of searching for a partitioning from $O(2^n)$ to $O(n \log n)$), **efficient** (for instance, the areas are precomputed only once, as seen in line 71 of our code), and **effective** (as demonstrated by the experiments in Figure 5).
> > > > Thus, while it may not be theoretically interesting, from our perspective, it offers engineering advantages akin to those simple and effective methods such as LPA and PageRank.
> > > > Even though this method has unfortunately left you an unfavorable impression of being ad hoc and heuristic, we think this comes from its **intuitiveness** and do not necessarily view this as a flaw.
> > > >
> > > > By all means, we hold your opinions in high regard and will continue to delve into the efficiency improvements of NT in our future work.
> > > > Please do not hesitate to express any of your personal concerns.

---

> > > > > ### Comment · Reviewer_zjDW · 2024-11-28
> > > > >
> > > > > I thank your further response, and I have a few comments regarding your point of view.
> > > > >
> > > > > **For W2.3** You used an example '{Evidence 1, Evidence 2} - Agent 1 - Conclusion - Agent 2 - {Evidence 3, Evidence 4}' and mentioned that GPRGNN will aggregate the messages like '{Evidence 1, Evidence 2, Evidence 3, Evidence 4} - Conclusion' in the 2-hop perspective. If I understand correctly, you want to show that **GPRGNN's weight to the two-hop neighbors is uniform (identical)**. If so, I agree. However, **there have been many adaptive GNNs in recent years that can assign non-uniform weights to 2-hop neighbors**, such as ACM-GNN [1], which is mentioned in my original review, FAGCN [2], ALT [3], and most GT works such as [4]. I think this problem is regarding expressiveness, and if the proposed method truly has an advantage in aggregating messages from 2-hop neighbors, **a rigorous theoretical result would be appreciated, compared to [1-4] and even more works**. I understand this might be challenging, but to be frank, as I mentioned in my original review, **many existing works could assign high (but non-identical) weights to 2-hop neighbors; this paper does not have a very unique theoretical advantage**, at least based on my current understanding of the proposed method.
> > > > >
> > > > > **For W4** Thank you for your response. I agree that this might be a matter of taste. Again, my personal evaluation is that Section 4.2 is ad hoc and heuristic-based. If most other reviewers and AC do not think so, I would be happy to see their high evaluation of this section.
> > > > >
> > > > > [1] Luan, Sitao, et al. "Revisiting heterophily for graph neural networks." Advances in neural information processing systems 35 (2022): 1362-1375.
> > > > >
> > > > > [2] Bo, Deyu, et al. "Beyond low-frequency information in graph convolutional networks." Proceedings of the AAAI conference on artificial intelligence. Vol. 35. No. 5. 2021.
> > > > >
> > > > > [3] Xu, Zhe, et al. "Node classification beyond homophily: Towards a general solution." Proceedings of the 29th ACM SIGKDD Conference on Knowledge Discovery and Data Mining. 2023.
> > > > >
> > > > > [4] Rampášek, Ladislav, et al. "Recipe for a general, powerful, scalable graph transformer." Advances in Neural Information Processing Systems 35 (2022): 14501-14515.

---

> > > > > > ### Author Response · Authors · 2024-12-02
> > > > > > **The difference among NT, signal-based GNNs, and multi-hop GNNs**
> > > > > >
> > > > > > We are grateful for your valuable recommendations.
> > > > > > Upon thorough examination of the literatures on ACM-GNN, FAGCN, and ALT, we have observed that these methods essentially involve a combination of low-pass information (propagated via $\epsilon \boldsymbol{I} + \boldsymbol{\tilde A}$) and high-pass information (propagated via  $\epsilon \boldsymbol{I} - \boldsymbol{\tilde A}$), as opposed to the conventional GNNs that solely utilize low-pass information.
> > > > > > To clarify, while low-pass filters can be understood as smoothing information between adjacent nodes, we interpret high-pass filters as mechanisms that drive the representations of connected neighbors apart from each other.
> > > > > > In heterophilic graph settings, the high-pass filter's effect leverages the dissimilarity among 1-hop neighbors, which is distinct from NT's strategy of leveraging the similarity among 2-hop neighbors.
> > > > > >
> > > > > > While we acknowledge that ACM-GNN, FAGCN, and ALT still primarily learn from 1-hop neighbors, methods such as GPRGNN and MAGNA [1] focus on learning from more distant neighbors.
> > > > > > However, these methods propagate messages along direct edges, leading to a compression of diverse information at intermediate nodes in heterophilic graphs.
> > > > > > For instance, in Figure 2a, the information from nodes 2, 3, 4, 5, and 6 is compressed at node 2.
> > > > > > Node 1 must then extract its relevant information, e.g. originating from node 3, from a message that has been mixed with node 2's dissimilar information.
> > > > > > Although these approaches can emphasize specific 2-hop neighbors, such as node 3, by assigning large edge weights along the path of 'node 3 - node 2 - node 1', this does not aid in disentangling the useful information of node 3 from the squashed messages from node 2.
> > > > > > In contrast, NT directly facilitates information exchange among 2-hop neighbors, thereby avoiding the issue of over-squashing.
> > > > > >
> > > > > > Moreover, when considering multi-hop message passing, the influence of a source node on a target node is constrained by certain factors [2], which may result in over-squashing, including the width $h$ of the intermediate node representations.
> > > > > > The simplified NT (as detailed in our Proposition 2 in the previous comments) can be regarded as an MPNN with intermediate representations $(\boldsymbol{K}_v, \boldsymbol{K}_1)$, which, when flattened, has a width of $h \times (h+1)$.
> > > > > > Consequently, NT is capable of alleviating the over-squashing issue, particularly in heterophilic graphs, with its wider representations when transmitting messages to distant nodes.
> > > > > >
> > > > > > > [1] Guangtao Wang, Rex Ying, Jing Huang, Jure Leskovec:
> > > > > > Multi-hop Attention Graph Neural Networks. IJCAI 2021: 3089-3096
> > > > > > >
> > > > > > > [2] Francesco Di Giovanni, Lorenzo Giusti, Federico Barbero, Giulia Luise, Pietro Lio, Michael M. Bronstein:
> > > > > > On Over-Squashing in Message Passing Neural Networks: The Impact of Width, Depth, and Topology. ICML 2023: 7865-7885

---

> > > > > > > ### Comment · Reviewer_zjDW · 2024-12-02
> > > > > > >
> > > > > > > I thank the authors for this discussion. I would be happy to see the above discussion more rigorously and formally in future revised versions (I know updating PDF is not allowed at this phase). Also, comparing the expressiveness between the proposed work and [1] should be interesting.
> > > > > > >
> > > > > > > [1] Rampášek, Ladislav, et al. "Recipe for a general, powerful, scalable graph transformer." Advances in Neural Information Processing Systems 35 (2022): 14501-14515.

---

> > > > > > > > ### Author Response · Authors · 2024-12-03
> > > > > > > > **The difference between NT and Graph Transformers**
> > > > > > > >
> > > > > > > > Thank you for your valuable feedback.
> > > > > > > >
> > > > > > > > In Related Works in our manuscript, we have discussed the topic of Graph Transformers (GT), with a specific focus on GraphGPS.
> > > > > > > > As we have summarized, GraphGPS and many other GTs comprise three key components:
> > > > > > > >
> > > > > > > > * **MPNN**: message passing neural networks, which encompass many conventional GNNs, including GCN, GAT, and GIN [1].
> > > > > > > > * **GT**: the self-attention applied to the entire node set.
> > > > > > > > * **PE/SE**: global/local positional/structural encodings.
> > > > > > > >
> > > > > > > > As is known, the expressiveness of MPNN is at most equivalent to the 1-WL test [1].
> > > > > > > > GT, by nature, treats the graph as fully-connected, which limits its ability when distinguishing graphs with an equal number of nodes, thus rendering it less expressive than the 1-WL test.
> > > > > > > > The superior expressiveness of GraphGPS is primarily attributed to PE/SE.
> > > > > > > > As discussed in Sections 3.2 and 3.4 of the literature on GraphGPS [2], the judicious use of PE/SE can assign discriminable colors to nodes in the WL test, thereby elevating GraphGPS to a universal approximator with greater expressiveness than any WL test.
> > > > > > > > Indeed, with strong PE/SE, even an MLP can achieve high expressiveness [3], surpassing any WL test.
> > > > > > > >
> > > > > > > > In conclusion, the vanilla GT is less expressive than MPNN and NT, but the integration of PE/SE can enhance their expressiveness (GT, MPNN, and NT) to a significant degree.
> > > > > > > >
> > > > > > > > We hope that this comparison with GraphGPS will meet your satisfaction.
> > > > > > > > At present, we are engaged in a formal analysis on the sensitivity bound [4] of NT, in the hope to catch up tomorrow's deadline.
> > > > > > > >
> > > > > > > > Best regards.
> > > > > > > >
> > > > > > > > > [1] Keyulu Xu, Weihua Hu, Jure Leskovec, Stefanie Jegelka: How Powerful are Graph Neural Networks? ICLR 2019
> > > > > > > > >
> > > > > > > > > [2] Ladislav Rampásek, Michael Galkin, Vijay Prakash Dwivedi, Anh Tuan Luu, Guy Wolf, Dominique Beaini: Recipe for a General, Powerful, Scalable Graph Transformer. NeurIPS 2022
> > > > > > > > >
> > > > > > > > > [3] Liheng Ma, Chen Lin, Derek Lim, Adriana Romero-Soriano, Puneet K. Dokania, Mark Coates, Philip H. S. Torr, Ser-Nam Lim: Graph Inductive Biases in Transformers without Message Passing. ICML 2023: 23321-23337
> > > > > > > > >
> > > > > > > > > [4] Francesco Di Giovanni, Lorenzo Giusti, Federico Barbero, Giulia Luise, Pietro Lio, Michael M. Bronstein: On Over-Squashing in Message Passing Neural Networks: The Impact of Width, Depth, and Topology. ICML 2023: 7865-7885

---

### Official Review · Reviewer_BNji · 2024-11-03

**Soundness:** 3
**Presentation:** 3
**Contribution:** 3
**Rating:** 8
**Confidence:** 4

**Summary:**

The paper presents Neighbourhood Transformers (NT), a new model to leverage the monophily for graph learning. NT uses self-attention to share information among nodes and collect data from their broader neighborhoods, focusing on nodes more similar to their 2-hop connections. Besides, NT addresses the problem of high memory and time demands due to different neighborhood sizes.

**Strengths:**

-The introduction of Neighbourhood Transformers (NT) based on the monophily assumption can be viewed as a contribution to graph learning, and its capacity for expansion appears to be advantageous .

-The paper is generally structured clearly and well-written.

-Experimental results show that the proposed model has comparable or improved performance.

**Weaknesses:**

-It is recommended to discuss related work focusing on monophily.

-Some aspects of the experimental analysis lack depth. For instance, it is observed that the performance improvement of NT on homophilic graphs is not as significant as on heterogeneous graphs, yet no in-depth analysis is provided.

**Questions:**

-Since linear attention part in NT is inspired by existing work, is there has any limitations of the linear attention mechanism , and how might these affect the model's performance? Are there specific scenarios where linear attention might fall short compared to traditional self-attention?

-How does the NT perform on graphs with extreme degree distributions that are significantly different from those in the training?

---

> ### Author Response · Authors · 2024-11-15
> **Reply to the Weaknesses and Questions**
>
> Dear Reviewer,
>
> Thank you for your patience and for providing us with the opportunity to address your concerns.
> We have carefully considered your comments and attempt to resolve them as follows.
> Should you find any issues or omissions in our responses, please do not hesitate to inform us.
>
> ## W1: It is recommended to discuss related work focusing on monophily.
>
> We acknowledge your point regarding the monophily.
> In fact, monophily is a property that emerges from heterophily.
> When a graph is heterophilic, where the traditional message passing methods for homophily fails, we may further utilize the property of monophily to help handling heterophily.
> Thus, the work of monophily is actually a subset of the work of heterophily.
> In our Introduction and Related Works sections, we have discussed heterophily, which inherently covers the subset of monophily-related work.
>
> ## W2: Need more experimental analysis such as why the performance improvement of NT on homophilic graphs is not as significant as on heterogeneous graphs.
>
> We appreciate your suggestion for further experimental analysis.
> On heterophilic graphs, different from baselinses' utilizations of the plain second-order adjacencies, NT exchanges messages conditioned on the central node, which enriches the diversity of patterns and enhances representation expressiveness, as detailed in lines 100 to 107.
> On homophilic graphs, NT may downgrade to 'mimic' traditional message passing, which is already effective for such graphs, as explained in lines 451 to 453 and 223 to 225.
> In conclusion, NT has an improvement when comparing against heterophilic GNNs while are compatible with homophilic GNNs.
> We think this is why NT's performance improvement on heterophilic graphs is more significant than on homophilic graphs.
>
> ## Q1: Is there any limitation of the linear attention mechanism? How might these affect the model's performance? Are there specific scenarios where linear attention might fall short compared to traditional self-attention?
>
> We have conducted experiments to investigate the limitations of the linear attention mechanism, as detailed in lines 522 to 530 and illustrated in Figure 6.
> Our findings indicate that while a pure linear-attention model (Performer) underperforms on 5 out of 10 datasets, our NT method, which is a hybrid of linear-attention and self-attention (Switchable), remains competitive with pure self-attention (Transformer) across all datasets.
> This suggests that the combined approach mitigates the potential shortcomings of linear attention in certain scenarios.
>
> ## Q2: How does the NT perform on graphs with extreme degree distributions that are significantly different from those in the training?
>
> Your question regarding the performance of NT on graphs with extreme degree distributions is insightful.
> We have conducted an analysis to measure the discrepancy between degree distributions in the training set and beyond.
> We first approximate the averaged size of belonging neighbourhoods for each node using $S = \text{deg}(A^2) / \text{deg}(A)$, where A is the adjacency matrix and deg is a function to derive node degrees from the adjacency matrix.
> Then, with 100 histogram bins, we transform $S[m]$ and $S[\bar m]$ into two distributions $P$ and $Q$, respectively, where $m$ indicates the training set.
> After that, we calculate the discrepency between the two distributions as $\sum\limits_i |p_i - q_i|$ and report the discrepencies for the 10 datasets in the following table.
>
> |              | Descrepency |
> |--------------|-------------|
> | Roman Empire | 3.5%        |
> | A-ratings    | 5.3%        |
> | Minesweeper  | 0.6%        |
> | Tolokers     | 8.8%        |
> | Questions    | 3.9%        |
> | A-computer   | 6.5%        |
> | A-photo      | 8.8%        |
> | CoauthorCS   | 6.5%        |
> | CoauthorPhy  | 7.2%        |
> | WikiCS       | 14.4%       |
>
> The results indicate varying levels of discrepancy across the datasets, with Minesweeper showing low discrepancy and WikiCS showing high discrepancy.
> Despite these variations, NT maintains consistent performance, as demonstrated by the experiments in our paper.
>
> We hope that these responses adequately address your concerns and provide a clearer understanding of our work.
> We are committed to further refining our manuscript based on your valuable feedback.
>
> Best regards.

---

### Official Review · Reviewer_GZCm · 2024-11-04

**Soundness:** 2
**Presentation:** 3
**Contribution:** 2
**Rating:** 5
**Confidence:** 4

**Summary:**

The paper proposes a neighborhood transformer approach to encode the monophily characteristic, which has been shown to be crucial to handle heterophilic graphs. With transformer applied to the neighborhood of all nodes, a neighborhood partitioning algorithm is designed to reduce the memory footprint of the quadratic transformer costs. Experiments show the better performance on both homophilic and heterophilic graphs than the GNN-based approaches.

**Strengths:**

1. The proposed method is designed to encapsulate the monophilic property of the graph benchmarks by neighborhood transformer, which shows good improvements compared to the GNN-based approaches.
2. The efficiency and memory footprint of the proposed model are further optimized by leveraging linear transformer (i.e., Performer) and designing neighborhood partitioning algorithms based on neighborhood size and area.

**Weaknesses:**

1. The neighborhood transformer is not quite scalable to handle large-scale graphs given the fact that it requires to apply transformers to the neighborhood of all nodes. Despite the partitioning algorithms, the computations are still intense.
2. There exist a lot of efficient graph transformers which can handle million-scale graphs, which perform well even on heterophilic graphs, e.g., NAGphormer, VCR-Graphormer etc.
   - From the scalability perspective, the proposed approach was only tested on the graphs with 40k node at most which is not sufficient to demonstrate the optimization from its partitioning algorithms.
   - From the evaluation perspective, the paper does not compare with any graph transformer baselines which have been shown to be the SOTA on quite a few benchmarks.
3. Equations 1~3 should be reversed to correctly show the computation orders step-by-step.
4. Technically, the novelty and soundness of this work is insufficient. To me, the major contribution is the partitioning algorithm which does not seem to be quite neat. E.g., for the size based partitioning, how is the p determined? For the area based partitioning, what are the pros and cons compared to those classic graph partitioning algorithms (e.g., METIS) and sparsification algorithms?

**Questions:**

1. For the size based partitioning, how is the p determined?
2. For the area based partitioning, what are the pros and cons compared to those classic graph partitioning algorithms (e.g., METIS) and sparsification algorithms?
3. Does the proposed approach perform better than the SOTA graph transformers?

---

> ### Author Response · Authors · 2024-11-15
> **Explaining on the partitioning algorithm**
>
> Dear Reviewer,
>
> We sincerely appreciate your valuable insights and the time you have dedicated to reviewing our manuscript.
> Upon careful consideration of your comments, we have noted that there might be some misunderstandings regarding the partitioning algorithm we have proposed.
> We would very like to address these concerns (**Q2 & Q1**) first, in order to provide a clearer understanding of our work and facilitate a more accurate assessment.
> Following this clarification, we will proceed to address the other issues you have raised, such as the SoTA Graph Transformers (**Q3**), the novelty (**W4**), and the scalability (**W1 & W2**) of our approach.
>
> ## Q2: What is the difference between the partitioning algorithm and METIS-like algorithms?
>
> We would like to clarify a point regarding our partitioning algorithm.
>
> The primary objective of the algorithm is not to partition a large graph into independent subgraphs for mini-batch training, akin to the METIS algorithm.
> Instead, our algorithm groups graph nodes in a way that still requires full-batch training.
> This is because, in NT, applying transformers in all neighborhoods is only an intermediate step; the final representation of a node is then aggregated from all its belonging neighborhoods.
>
> We illustrate this difference with an example.
> Suppose METIS divides a graph into isolated subgraphs A, B, and C.
> A graph model can then be applied to one of these subgraphs, say A, to obtain the representation of a node, say $v$ that is in subgraph A.
> However, in NT, even if we divide graph nodes into groups A, B, and C, the process is different.
> If node $v$ is in group A and has connected nodes $u$ in group B and $w$ in group C, the representation of node $v$ is not derived from applying the transformer in group A.
> Instead, it is obtained by gathering information from the neighborhoods of nodes $u$ and $w$ after the transformer has been applied to groups B and C, respectively.
> In summary, while the graph nodes are partitioned in NT, all their neighborhoods must still be processed by the transformer, with their results being combined later.
>
> Our partitioning algorithm is designed to optimize both memory and computation time during the transformer's processing, before the results combination, as depicted in Figure 1 and explained in lines 73 to 80 of our manuscript.
> Thus, our method is tailored only to address the unique challenges that arise in NT and is fundamentally different from METIS-like graph partitioning algorithms.
>
> We sincerely hope that this explanation clarifies any misunderstanding you may have had about our partitioning algorithm.
> Should there be any further lack of clarity, please do not hesitate to reach out to us.
>
> ## Q1: How is $p$ determined?
>
> As mentioned in line 195 of Section 3.3, $p$ is a hyperparameter coming from Performer [1].
> The original article on Performer indicates that setting $p = h \log h$ is suffices to accurately approximate the attention matrix.
>
> > [1] Krzysztof Marcin Choromanski, et al.: Rethinking Attention with Performers. ICLR 2021

---

> ### Author Response · Authors · 2024-11-15
> **Q3: Should compare performance with SotA graph transformers.**
>
> We would like to provide a detailed comparison between our proposed NT and the SoTA Graph Transformers (GT), specifically the Polynormer [1], by referencing the latest data from Polynormer’s publication.
> We have compiled the following two tables to illustrate this comparison.
>
> **Table A: Performance on heterophilic graphs.**
>
> |            | Roman Empire   | A-ratings      | Minesweeper    | Tolokers       | Questions      |
> |------------|----------------|----------------|----------------|----------------|----------------|
> | GraphGPS   | 82.00±0.61     | 53.10±0.42     | 90.63±0.67     | 83.71±0.48     | 71.73±1.47     |
> | NAGphormer | 74.34±0.77     | 51.26±0.72     | 84.19±0.66     | 78.32±0.95     | 68.17±1.53     |
> | Exphormer  | 89.03±0.37     | 53.51±0.46     | 90.74±0.53     | 83.77±0.78     | 73.94±1.06     |
> | NodeFormer | 64.49±0.73     | 43.86±0.35     | 86.71±0.88     | 78.10±1.03     | 74.27±1.46     |
> | DIFFormer  | 79.10±0.32     | 47.84±0.65     | 90.89±0.58     | 83.57±0.68     | 72.15±1.31     |
> | GOAT       | 71.59±1.25     | 44.61±0.50     | 81.09±1.02     | 83.11±1.04     | 75.76±1.66     |
> | Polynormer | *92.55±0.37*   | **54.81±0.49** | **97.46±0.36** | **85.91±0.74** | **78.92±0.89** |
> | **NT**     | **94.77±0.31** | *54.25±0.50*   | *97.42±0.50*   | *85.69±0.54*   | *78.46±1.10*   |
>
> In Table A, we observe that NT outperforms GT on the Roman Empire dataset by a significant margin due to its utilization of directional information.
> Since GT layers do not consider the edges of nodes, they are unable to model the directionality of edges.
> On other heterophilic datasets, NT is only slightly behind Polynormer.
>
> We would like to emphasize that the hyperparameter budgets for Polynormer are higher than those for NT.
> Specifically, even without considering the additional GT layers, the maximum number of its convolutional layers is 10, while for NT, it is 5.
> This is because many of our (and also Performer's) baselines, which are cited from [2], are with this low-budget settings.
>
> **Table B: Performance on homophilic graphs.**
>
> |            | A-computer     | A-photo        | CoauthorCS     | CoauthorPhy    | WikiCS         |
> |------------|----------------|----------------|----------------|----------------|----------------|
> | GraphGPS   | 91.19±0.54     | 95.06±0.13     | 93.93±0.12     | 97.12±0.19     | 78.66±0.49     |
> | NAGphormer | 91.22±0.14     | 95.49±0.11     | *95.75±0.09*   | **97.34±0.03** | 77.16±0.72     |
> | Exphormer  | 91.47±0.17     | 95.35±0.22     | 94.93±0.01     | 96.89±0.09     | 78.54±0.49     |
> | NodeFormer | 86.98±0.62     | 93.46±0.35     | 95.64±0.22     | 96.45±0.28     | 74.73±0.94     |
> | DIFFormer  | 91.99±0.76     | 95.10±0.47     | 94.78±0.20     | 96.60±0.18     | 73.46±0.56     |
> | GOAT       | 90.96±0.90     | 92.96±1.48     | 94.21±0.38     | 96.24±0.24     | 77.00±0.77     |
> | Polynormer | **93.68±0.21** | **96.46±0.26** | 95.53±0.16     | 97.27±0.08     | **80.10±0.67** |
> | **NT**     | *92.61±0.63*   | *96.12±0.39*   | **96.07±0.32** | *97.32±0.11*   | *80.04±0.61*   |
>
> In Table B, NT is ranked as one of the top 2 methods across all five homophilic datasets.
> The averaged rank for NT is $(2+2+1+2+2)/5=1.8$, which is better than Polynormer’s average rank of $(1+1+4+3+1)/5=2$.
> This demonstrates that our method maintains strong competitiveness when compared to SoTA GTs.
>
> In conclusion, NT demonstrates competitive performance against the SoTA GTs across both heterophilic and homophilic datasets.
>
> However, we would like to emphasize additionally that, as analyzed in the Related Works, GTs are unable to replace Message Passing Neural Networks (MPNN) as an independent method due to the information loss of topology.
> Consequently, the current GT methods are usually combined with MPNNs (e.g. GraphGPS) and would be more accurately described as 'GT-augmented MPNNs'.
> In contrast, our research demonstrates that NT is capable of handling heterophily and can be adaptively compatible with MP, potentially replacing it as a key component in future GNN architectures.
> This is actually orthogonal to the GT approach; that is, NT can also be combined with GT to form an enhanced 'GT-augmented NT'.
> This is the primary reason we did not directly compare GT and NT in the main text of the manuscript.
>
> We will include the above comparisons in the appendix of our paper to provide a more comprehensive evaluation of our method.
> Thank you again for your valuable suggestions.
>
> > [1] Krzysztof Marcin Choromanski, et al.: Rethinking Attention with Performers. ICLR 2021
> >
> > [2] Oleg Platonov, et al.: A critical look at the evaluation of GNNs under heterophily: Are we really making progress? ICLR 2023
> >
> > [3] Ladislav Rampásek, et al.: Recipe for a General, Powerful, Scalable Graph Transformer. NeurIPS 2022

---

> > ### Author Response · Authors · 2024-11-15
> > **The reply for your concerns in Weaknesses**
> >
> > Except for the order of Equations 1~3 (**W3**), which we will adjust accordingly, and those already answered in previous questions, we are committed to addressing your concerns regarding the scalability and novelty of our work.
> >
> > Below, we provide detailed responses to your points.
> >
> > ## W4: On the novelty
> >
> > Building on our previous responses to **Q1** and **Q2**, we hope you can recognize the research value of the following contributions of our work:
> >
> > **First**, we introduce a novel approach that focuses on information exchange within neighborhoods.
> >
> > While extensive research has been conducted on two extreme styles of message passing, traditional edge-wise passing and the transformer’s edge-less passing, there is a vast area of research potential between these styles that has not been fully explored.
> > As illustrated in Figure 2 of our paper, we strongly believe that our NT, as an intermediate passing style with demonstrated advantages, offers significant benefits to the community.
> >
> > **Second**, we successfully validate the effectiveness of NT with our innovative partitioning algorithm.
> >
> > The simplicity of the NT idea, despite its effectiveness, raises the question of why it has not been studied in the years following the introduction of GCN (2016) and GT (2020).
> > We argue that the primary challenge lies in the memory and computational complexity of NT.
> > As Section 5.2 of our paper shows, without our partitioning algorithm, the number of datasets on which NT can be run is very limited, even with high-end hardware like the NVIDIA A800 GPU card (with 80GB memory).
> > Our algorithm, however, enables the validation of this idea across all 10 homophilic and heterophilic graphs on a GPU card with just 4GB of memory, bringing NT from an idea into emperical practice.
> >
> > ## W1 & W2: On the scalability
> >
> > Our response to **Q1** confirms that NT is trained in full-batch mode.
> > Consequently, your concern about scalability is well-founded: the current implementation of NT is not optimized for large-scale graphs.
> >
> > We agree that scalability is crucial for the broader application of NT, and we are keen to explore stochastic training methods for NT in a mini-batch setting.
> > For this paper, we believe that demonstrating our claims on 10 moderate-scale graphs with varying degrees of homophily is sufficient.
> > We propose to leave the exploration of scalability for large-scale graphs to future work.
> > This is analogous to the development of GraphSAGE following GCN and NAGphorer following vanilla GT.
> > We understand that decoupling the method itself from its efficiency improvements may be a subjective decision, and we are more than willing to engage in further discussions on this matter in the coming days.
> >
> > We appreciate your attention to these aspects of our work and look forward to your feedback.
> >
> > Warm regards.

---

> > > ### Comment · Reviewer_GZCm · 2024-11-25
> > >
> > > Thank you for providing the detailed responses, which have addressed most of my concerns raised in the initial reviews. But how much improvement of this proposed approach compared to existing graph transformers is still limited (as the authors also answered in the reply). I have updated my score accordingly.

---

> > > > ### Author Response · Authors · 2024-11-26
> > > > **Thank you for your response**
> > > >
> > > > We are delighted to learn that you are almost satisfied with the changes we have made and are very encouraged by the improved score.
> > > >
> > > > By the way, we would like to sincerely apologize that we have inserted into our comments earlier with an additional paragraph about GTs, which might not have caught your attention.
> > > > We cite the added paragraph below to forestall this contingency:
> > > >
> > > > > However, we would like to emphasize additionally that, as analyzed in the Related Works, GTs are unable to replace Message Passing Neural Networks (MPNN) as an independent method due to the information loss of topology. Consequently, the current GT methods are usually combined with MPNNs (e.g. GraphGPS) and would be more accurately described as 'GT-augmented MPNNs'. In contrast, our research demonstrates that NT is capable of handling heterophily and can be adaptively compatible with MP, potentially replacing it as a key component in future GNN architectures. This is actually orthogonal to the GT approach; that is, NT can also be combined with GT to form an enhanced 'GT-augmented NT'. This is the primary reason we did not directly compare GT and NT in the main text of the manuscript.
> > > >
> > > > In brief, we believe that a direct comparison between NT and GT is a little bit inappropriate because GTs are essentially 'GT + MP', whereas our paper aims to demonstrate that 'NT > MP', particularly in handling heterophily.
> > > >
> > > > We hope this clarification can address your concerns regarding the comparison with GTs.
> > > > Thank you for your consideration of our revisions and the efforts we have invested.

---

### Author Response · Authors · 2024-11-21
**A new manuscript PDF is uploaded**

Dear Reviewers,

We have updated our manuscript PDF in accordance with your valuable comments.
The changelog is as follows:

* An anonymous URL to our experimental code is updated in the Abstract, as R3 suggested in W5
* The orders of Equations in line 165, line 176, lines 191-194, and lines 205-208, are reversed as R1 suggested in W3
* The edge attributes "M'" in Eq.3 (now it is Eq.1 due to the previous change) is removed to simplify the article since "M'" never appears in our experiments
* The comparison with SotA graph transformers is appended to Appendices C, as responded to R1's Q3
* Analysis on the degree distribution descrepency is appended to Appendices D, as responded to R2's Q2
* New baselines (CDE and BloomGML) are added to Table 1 as R3 suggested in W3
* Theoretical analysis on the expressiveness is appended to Appendices E, as suggested by R3 in W1
* Theoretical analysis on the partitioning algorithm is appended to Appendices F, as suggested by R3 in W4

We are immensely grateful for your suggestions, which have significantly contributed to strengthening our work.
Also, please allow us to kindly ask for your attention in reviewing our responses before the rebuttal deadline.
Should there be any omissions or insufficient explanations on our part, please inform us promptly.
We are honored to have this opportunity to communicate with you once again.

Sincerely.

---

### Comment · Area_Chair_Dpr7 · 2024-11-23

Dear Reviewers,

The authors have uploaded their rebuttal. Please take this opportunity to discuss any concerns you may have with the authors.

AC

---

### Author Response · Authors · 2024-11-27
**PDF changelog**

Dear Reviewers,

We have revised our manuscript PDF with some minor term fix in accordance with the W1 of Reviewer zjDW.

Warm regards.

---

### Meta-Review · Area_Chair_Dpr7 · 2024-12-21

**Metareview:**

The paper introduces the Neighbourhood Transformer (NT), a model designed to leverage the monophily assumption for graph learning by using self-attention on node neighborhoods. NT incorporates a neighborhood partitioning algorithm to address the high memory and computational demands associated with transformer models. Experimental results demonstrate that NT performs better than GNN-based approaches, particularly on heterophilic graphs.

While the proposed NT model shows promising results, the reviewers have identified several weaknesses that need to be addressed:

1. The method, which is based on monophily information, uses 2-hop information to compute attention. However, several previous methods have already incorporated 2-hop information to address challenges related to heterophily, which somewhat reduces the novelty of this approach.

2. Despite using Performer and the partitioning algorithm, the method’s scalability remains a concern. The current implementation of the NT is not optimized for large-scale graphs. Scalability, particularly for node-level tasks, is crucial.

3. Compared to existing graph transformers, the improvement offered by the proposed approach appears limited, as it does not demonstrate significant advancements over previous methods.

Based on these weaknesses, we recommend rejecting this paper. We hope this feedback helps the authors improve their paper.

**Additional Comments On Reviewer Discussion:**

The authors have made several updates to their manuscript in response to reviewer feedback. They provided an anonymous URL to their experimental code in the Abstract, reversed the order of equations as suggested, and removed an unnecessary edge attribute from Eq. 3. They also added a comparison with state-of-the-art graph transformers in the appendices, included an analysis of degree distribution discrepancies, introduced new baselines (CDE and BloomGML) in Table 1, and provided theoretical analyses on the expressiveness of their model and the partitioning algorithm.

However, during the rebuttal and discussion phase, the reviewers’ concerns regarding novelty of the proposed method and scalability issue remain unresolved. As a result, I recommend rejection based on the reviewers’ feedback.

---

### Decision · Program_Chairs · 2025-01-22

Reject